# South Arabia's prehistoric monument landscape shows social resilience to climate change

Joy McCorriston[1]*, Lawrence Ball[2¤a], Michael J. Harrower[3], Ian M. Hamilton[2,4], Sarah J. Ivory[5], Matthew J. Senn[1¤b], Tara Steimer-Herbet[6], Abigail F. Buffington[1¤c], 'Ali Ahmad Al-Kathiri[7], 'Ali Musalam Al-Mahri[7]

1 Department of Anthropology, The Ohio State University, Columbus, Ohio, United States of America, 2 Department of Ecology, Evolution & Organismal Biology, The Ohio State University, Columbus, Ohio, United States of America, 3 Department of Near Eastern Studies, Johns Hopkins University, Baltimore, Maryland, United States of America, 4 Department of Mathematics, The Ohio State University, Columbus, Ohio, United States of America, 5 Department of Geosciences and the Earth and Environmental Systems Institute, Penn State University, State College, Pennsylvania, United States of America, 6 Laboratory of Prehistoric Archaeology and Anthropology, Université de Genève, Geneva, Switzerland, 7 Ministry of Heritage and Tourism, Salalah, Sultanate of Oman

¤a Current Address: Kent Wildlife Trust, Tyland Barn, Chatham Road, Sandling, Maidstone, Kent, United Kingdom
¤b Current Address: New York State Department of Transportation, Albany, New York, United States of America
¤c Current Address: Chronicle Heritage, Al-Ula, Kingdom of Saudi Arabia
* mccorriston.1@osu.edu

## Abstract

In arid regions across northern Africa, Asia and Arabia, ancient pastoralists constructed small-scale stone monuments of varying form, construction, placement, age, and function. Classification studies of each type have inhibited a broader model of their collective and enduring role within desert socio-ecosystems. Our multivariate analysis of 371 archaeological monuments in the arid Dhofar region of Oman identifies environmental and cultural factors that influenced variable placement and construction across a 7000-year history. Our results show that earlier monuments were built by larger, concurrent groups during the Holocene Humid Period (10,000–6000 cal BP). With increasing aridification, smaller groups constructed monuments and eventually switched to building them in repetitive visits. Our model emphasizes the core role of monuments as a flexible technology in social resilience among desert pastoralists.

## Introduction

Difficult access in modern deserts and narrow typological foci in regional studies have hindered broad empirical models of how pastoralists built and used archaeological monuments [1]. Chronological and spatial organization of desert monuments reflect the dynamic relationships among pastoralists and the arid ecosystems they embraced [2–4]. Pastoralists have persisted in the world's arid regions even as

**Data availability statement:** All the data are included in the paper with Supplemental Files. The same data have been placed also in an online data repository (tDAR-- the Digital Archaeological Record). This data is "Public" Here is the URL: https://core.tdar.org/search/results?_tdar.searchType=simple&query=ASOM

**Funding:** This research has been funded by the US National Science Foundation under the Coupled Human - Natural Systems Large Grants (CNH-L 1617185) (PI-JMcC); and the US National Science Foundation Human Social Dynamics Program (DHB 0624268) (PI-JMcC); funds contributing to field work came from a National Geographic Explorer Grant # EC-44704R-18 (PI-AFB) and the Ministry of Heritage and Tourism, Sultanate of Oman (J.McC, A.A.Al-K, A.Al-M). The funders had no role in study design, data collection and analysis, decision to publish, or preparation of the manuscript.

**Competing interests:** The authors have declared that no competing interests exist.

agricultural oases emerged, flourished, and disappeared. Across Arabia, multiple formal classes of small-scale stone monuments correspond to cultural, chronological, and functional types [5–20]. Spatial and temporal distributions of ancient monuments serve as proxies for ancient human economic and social activities in diverse environments [21–23]. In Arabia, changes in monument types took place against a backdrop of persistent pastoralism across major Holocene climate changes [24]. As the landscape grew more arid, the biomass and seasonal availability of forage to support animals and herders declined [25,26]. Albeit from few studies, Arabian faunal remains also suggest that cattle herders thrived in wetter environments in the early to mid-Holocene and adjusted their focus to different taxa according to available graze and browse as climates changed [27–29] so that cattle herders preferred wetter environments than today's. In Dhofar's steep environmental gradient, cattle herders today remain in the forested mountains and narrow plateau near the coast [30]. The arid interior is today sparsely populated by mobile goat and camel herders. We describe how measurable changes in the archaeological record of monument construction and placement reflect the social resilience of a declining or more thinly dispersed population in arid landscapes and livestock replacement of cattle by caprines and camels.

Our study complements recent studies of archaeological cultures and climate proxy records that consider the impact of aridification phases on prehistoric populations [21,31,32]. Diverse cultures in diverse circumstances practiced different strategies for resilience [33]. These include changes that are manifest as, for example, changes in settlement and mobility, shifting resources and adopting new ones, engineering flexible field and water catchment systems, adjusting exchange networks, and employing traditional ecological knowledge and cosmology in a changing world [34–37]. Some strategies manifest truly transformative resilience, such as the settlement of oases and inception of Arabian agricultural systems. Other strategies reflect an adaptive social resilience, manifest in the changing styles of Arabian monuments.

Based on excavated and dated materials, Arabian monuments appeared in sequences of different forms for different cultural groups [38]. In Dhofar, Oman, Neolithic monuments (7500–6200 cal BP) are sub-circular platforms; Bronze Age monuments are tombs (High Circular Tombs or HCT, 5200–4000 cal BP); Iron Age (2300–1700 cal BP) triliths take the form of elongate, low stands supporting triads of upright stones; and in Late Antiquity, clusters of boat-shaped graves (1100–750 cal BP) marked the passage of mobile people [14,39]. (Fig 1) Recent dates on stepped concentric alignments of boulders (SCABS) elsewhere show a multi-period use [40]. While cultural syntaxes differed, monuments served a common purpose to communicate social collectivity and institutions among people who were rarely cohabitants [41–42]. Collectively, this record indicates both a timeline of different types [14,42] and a persistent tradition of monuments shaping social resilience in a dynamic pastoral landscape.

The Holocene Humid Period was an interval of higher than modern rainfall in southern Arabia that lasted from ~10,000–6000 cal BP [43–44]. Ample regional geological and paleoclimatic evidence demonstrates a pluvial interval, with standing

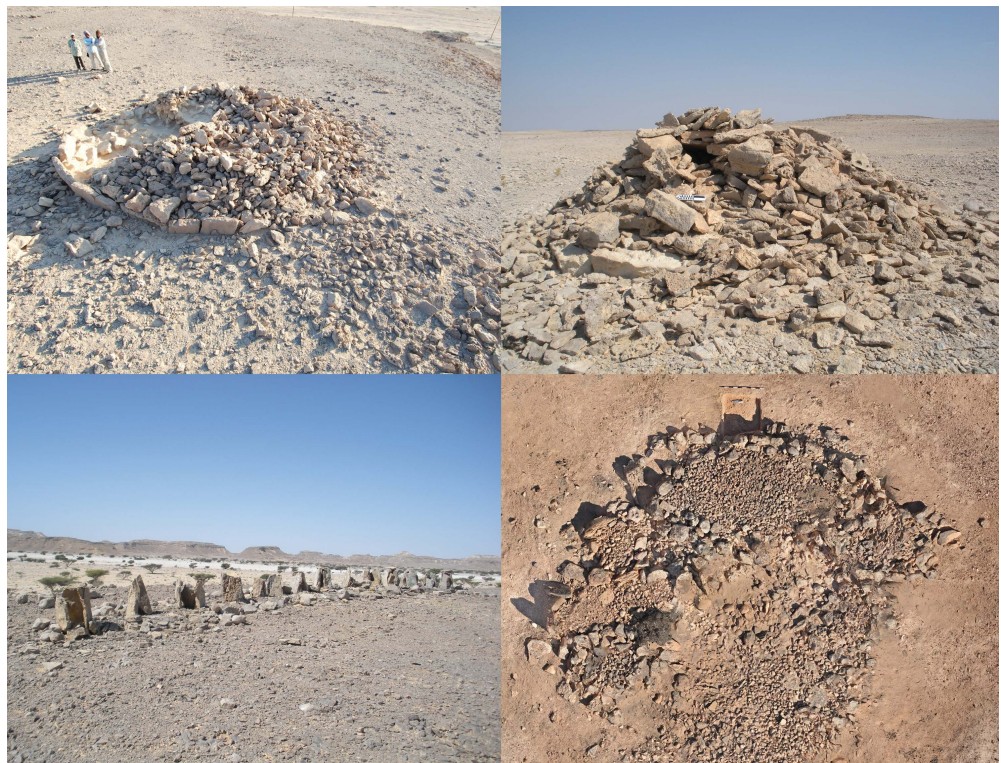

**Fig 1. Examples of major (by frequency) monument types from Dhofar.** Clockwise from upper left: Platform D028-001, meter scale visible in excavation trench near closest section; HCT D033-001, 20 cm scale visible at mid height; Boat graves D100-001 with meter scale at upper left; a trilith (unsurveyed) with uprights 40-50 cm height. Photo credits clockwise from upper left: Joy McCorriston, Jennifer Everhart, Wael AbuAzizeh, and Michael Harrower.

surface water in expansive inland paleolakes, and higher summer precipitation, particularly along the southernmost Arabian coast [25,45,46]. Mangroves, seasonal woodlands, and grasslands expanded, especially along the coasts and escarpments [44,47–50]. After 6000 cal BP, oxygen isotopes from speleothems suggest a gradual southward retreat of the ITCZ, resulting in a time-transgressive decline in rainfall [43], with drying likely to have occurred in a two-step manner[44]. After ~5000 cal BP, decreasing monsoonal rainfall resulted in the desiccation of most inland lakes. Finally, a period of intense aridity began around 2700 cal BP when local and regional records suggest maximum aridity for about a thousand years [43].

Using new data from Dhofar, Oman, we explore environmental and cultural factors influencing the placement of monuments, and we probe a correlation between climatic-environmental changes and changes in monument types. (Fig 2) Prior research suggests that large groups, perhaps in the thousands, convened episodically in the Neolithic [29,53]. Neolithic platforms cease to be constructed after the Holocene Humid Period, and several researchers have noted a correlation between Bronze Age tombs, oasis origins, and declining rainfall [24,26,54,55]. While some Arabian pastoralists may have adopted date-and-cereal cultivation in oases, pastoralists in Dhofar's deserts remained mobile with herds, and their monuments attest to their movements, offering insights into their adaptations to climate and environment.

## Materials and methods

### Monument survey methods

Lacking direct observations of ancient people, we rely on monuments as sources of proxy data on ancient pastoralists' behavior. Building on prior archaeological surveys across all of Dhofar [7,56], our team targeted a sample of monuments

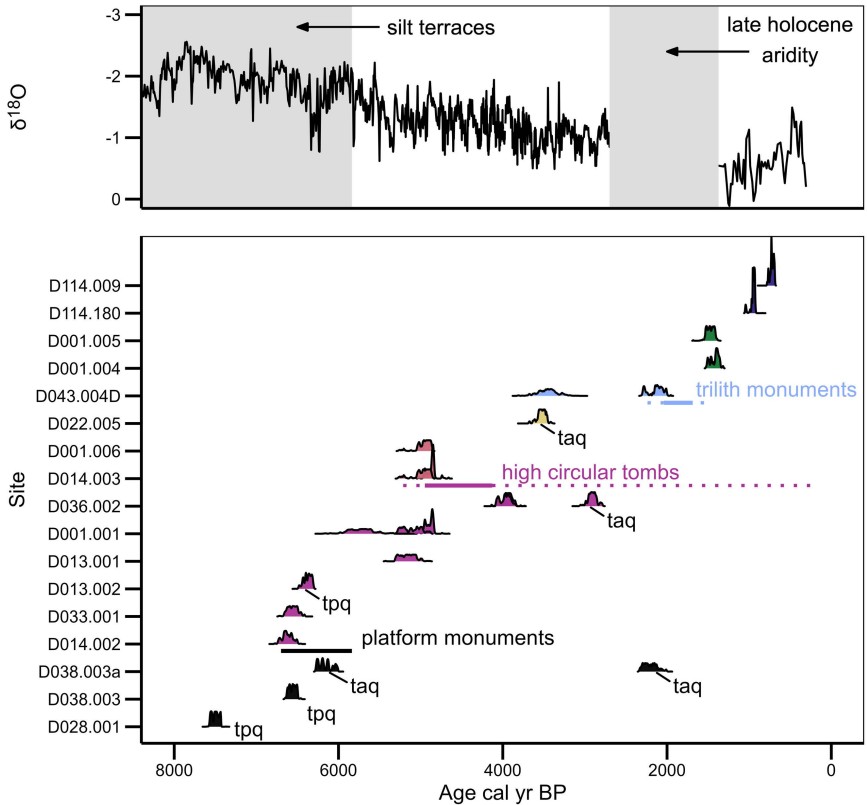

**Fig 2. Timeline of major monument types based on multiple radiocarbon determinations from excavated monuments in Hadramawt, Yemen and in Dhofar, Oman.** Top register compiles precipitation proxy record from Qunf Cave, Dhofar [51] with Holocene Humid Period proxy records (silt terraces) in Hadramawt [38]. For Dhofar, platforms (n=26, $^{14}$C=4,) are black, HCTs (n=135, $^{14}$C=10) are fuschia, triliths (n=165, $^{14}$C=2) are blue, and boat graves (n=22, $^{14}$C=2) are represented in dark purple, with other (infrequent) monument types (yellow, green, pink). Taq denotes "terminus ante quem," which indicates a stratigraphic position of sample that constrains the dated event to an earlier time; tpq denotes "terminus post quem," or an event dated after the radiocarbon determination. These priors define eras of monument construction and use. For example, Dhofar platforms appear **after 7500** cal yr. BP. Samples here are only from Dhofar [14,52]; radiocarbon calibrations generally match the Bayesian posteriors (horizontal lines) established with comparable data from Hadramawt, Yemen [12,16,38] and observations within the broader literature of Arabian prehistory. (See Supporting S1-Table, S2 Table, S3 Table for details). The long dotted line for HCT reflects stratigraphic evidence of re-use centuries and millennia after primary interments. Image by Shane Scaggs and Joy McCorriston.

and areas. We conducted a narrowed search within areas we knew to contain monuments as well as areas unknown to us. We adapted survey methods from prior research in a comparable terrain [16,57,58]. In 1996, McCorriston conducted a reconnaissance to Dhofar. Subsequently the team purchased high (0.6m) spatial resolution QuickBird satellite imagery for comprehensive survey. We selected imagery to cover a "T" with 11 of 14 images across the Nejd desert inland of the cloud-forested coastal mountains [59] p.146, Fig 1]. This selection targeted the region known to contain tombs and triliths and is comparable to terrain with these and other early monuments in other areas of Southern Arabia. A few images in the mountains and coastal plain ensured we would also sample there; we chose so that our broader results could be compared to others' [60]. In purchase of specific image blocks for comprehensive coverage—some with monuments, some unknown--we used prior knowledge and Google Earth to select locales accessible by roads and vehicle tracks.

Where possible, the team followed established vehicular-tracks both for logistics and to limit our footprint in a fragile ecosystem. Modern routes in the arid and deeply dissected terrain of Southern Arabia typically follow ancient tracks constrained by drainages and passes. We documented traces leading through areas frequented and shaped by pastoralists

and former caravans [30,61], something we documented during 22 years of fieldwork in Hadramawt and Dhofar. These routes may also have higher monument frequencies than off-route, difficult- to-access locales, which we examined by adding our foot tracks and our meanders to reach targets generated by a semi-automated detection of monuments [62]. Where the survey team's passage was rapid (i.e., major asphalt roads), we eliminated tracks from subsequent analysis; but we documented each monument we observed by a simple inventory list when time, resource and logistics prevented more thorough investigation. Survey over multiple years and projects nevertheless consistently collected all monument data within defined viewsheds (sampled areas).

Between 2009–2018, we surveyed 428 monuments. Opportunistic, judgmental sampling drawing on the expertise of crews with decades of collective field experience was complemented by collaboration with faculty and graduate students in engineering and statistics to develop algorithms that detected monuments in satellite imagery [62,63]. Our logged tracks became transects in which we recorded the presence and absence of monuments. This network of monuments and tracks served as the basis for viewshed analysis, a computation that maps the areas of the surrounding landscape visible from a given point or points based on whether those areas lie above the local landscape horizon [64,65]. We used an ASTER Global Digital Elevation Model (GDEM) (30 m resolution) and observer height set to 1.6 m to identify all areas visible from the locations we visited. Based on extensive field experience in which we measured maximum distance to sighted monuments, we set an 800 m viewshed boundary to define a full-survey coverage area.

In conformity with ethical standards, we conducted all field research under (un-numbered) permits issued annually through 5-year agreements signed between Ministry of Heritage and Tourism, Sultanate of Oman and Joy McCorriston (2007–2013, 2017–2021).

**Field documentation methods.** We documented monument attributes, including location, other visible monuments, materials, preservation, cultural type, and measurements to calculate stone volume and largest stones as proxies for large group sizes and simultaneous labor input [66]. We differentiate between *episodic monuments* (built in one episode) and *accretive monuments,* which can be assembled in repeated visits to a particular location. Our terminology refers to the initial construction of monuments according to a structural-cognitive template. We recognize, as both stratigraphic examples and in theoretical concept, that monuments were re-visited, altered, additions made after substantial lapses in time, sometimes partially or fully dismantled, and appropriated—such is the role of monuments in both shaping and being shaped by social groups as literal touchstones for institutional thought [41,67]. For example, through radiocarbon determinations, we documented instances of animal offerings placed into tombs opened and re-sealed a thousand years after a dated burial. This use is unrelated to initial construction, which was the target of our documentation through chronology and measurements.

Using high-precision GPS and tape measures, we established geographic coordinates on the center of a monument, its perimeter or length, and took compass bearings on orientation [59,62]. For all monuments, we measured the length of the largest stone, registered as a categorical variable greater or less than 0.25, 0.50, 0.75, 1.0 m. These ranges index labor group sizes from single to many people. For most small-scale stone monuments, the largest measured stone could be shifted by the two to four adults in one herding family, ethnographically averaging five to seven adults and children [30,68]. Our experience with excavation crews dismantling collapsed monuments in both Yemen and Oman, shows that to shift limestone blocks greater than 0.75 m length minimally requires seven or more adults, particularly as tools such as iron levers were unavailable in prehistory. Transport of such stones even a few meters implies the cooperation of a substantial group. We photographed each monument from multiple angles, which allowed us systematically to generate standardized estimates of stone volume from geometric shape and dimensions. We calculated the volume of stone using volumetric calculations for six geometric shapes, which closely match the monument shapes. Platforms can be cylinders as well as trapezoidal shape; SCAB monuments are cone-shaped; HCT are cylindrical x 0.67 to account for a hollow chamber; triliths are the shape of a triangular prism x 0.5 to account for empty space; and boat graves are straight-sided lozenges.

With excavations and radiocarbon assays in Dhofar, we refined an earlier typology of monuments [7,9,12,14] and identified details of construction. Subsequent data cleaning resulted in a data set of 371 surveyed monuments and the survey tracks. (S1-Table) Although fieldwork generated an initial array of 428 observations, some monuments were too poorly preserved to assign to a chronological-cultural type from surface inspection. For those that we could assign, we used the following visual criteria. Platforms have large blocks or upright slabs forming an outer base and an overall D-shaped or oval plan. Often the largest uprights form an eastern face. Platforms have a fill of smaller stones, and typically also have a (sometimes worked) standing stone set further to the east. (Excavations inside show that the uprights were supported by interior chock stones, in turn held in place by fill.) To balance and support stones and consolidate them into a free-standing, stable platform required rapid (i.e., episodic) initial construction. An HCT appears as a conical cairn 3–5 m across with a depressed center. This depression denotes an interior (usually collapsed) chamber made from upright or stacked slabs. Usually some of the chamber facing is still visible without excavation. Tomb (HCT) construction was surely also episodic, given the exigency of burial in a hot environment populated by scavengers and carnivores. A SCAB is also a conical structure, created by stacking boulders in concentric daises decreasing in diameter with the height of the monument. Triliths have clusters of 1–3 upright stones supported by a very low, oblong, gravel platform outlined by cobbles. It would be difficult to mistake this distinctive monument for any other type (Fig 1). Each trilith platform and its adjacent hearths is repeated along an alignment. We identified boat graves by their shape in plan and by the clustering of multiple, abutting, boat-shaped elements (Fig 1), each of which likely included a burial. In the cases of agglomerated triliths and boat graves, there is no construction characteristic that requires completion in a single episode. Triliths and boat graves could and sometimes did appear as isolated, unreplicated elements; this indicates that conglomerated examples were accretive, potentially completed over many years.

After analysis of excavated examples, it was possible to reassign some monuments from "unknown" and "tumulus" field designations. We reviewed original field notes and photographs for each monument, and these aided in revisions—based also on 10 years' experience--of other typological classifications. For some monuments, there were no measurements of stone size and volume, leading to their exclusion from our data. We also excluded rare types (e.g., Haluf tombs n = 3, Wall tombs n = 5) from analysis, narrowing the study to the five categories that represent 90 percent of documented monuments.

**Chronological and stratigraphic methods.** Altogether from Hadramawt and Dhofar we obtained a dated sample of 31 monuments (47 radiocarbon determinations) to complement existing literature. The number of radiocarbon determinations falls far short of reasonable sample sizes for summed probability frequency distributions, which are problematic even in large samples [69,70]. Instead, we note that our Bayesian analysis in Hadramawt using regional geomorphology priors there provides a framework consistent with the Dhofar radiocarbon determinations [38].

Radiocarbon determinations come from excavated samples with varying potential to date the initial construction of a monument. Sometimes dated samples are from bone (human and faunal) and determine when an organism died. Other samples are carbon from charcoal, buried humus, or shell. We used stratigraphic associations to contextualize how the deposition of such samples relates to the actual event of monument construction. With multiple monuments of each type, we documented construction materials and techniques, preservation, use, and re-use, including subsequent burials and sacrifices, evidence of collapse, additions, and subsequently quarried stone [16,52,71–73]. The probability determinations in Fig 2 represent events prior or posterior to monument construction. Also, as a responsible reporting of results, we include a radiocarbon determination we think has unexplained error (D033-001). Another determination (D014-002) also lies outside our expected range. (For stratigraphic details see [14,52]). More samples would surely refine and constrain our chronology, but we caution that excavation is a costly and sometimes unsuccessful path to obtain contextualized radiocarbon samples. Nevertheless, excavation remains the only viable strategy for a classification anchored in absolute dates.

Our classification and chronology largely correlate with observations across Arabia [3,13,15,17–20,34,39,40,74]. While multiple researchers have—like us—found episodes of re-use in existing monuments, we focus on the pattern of

initial construction our studies of stone placements, arrangements, inclusions, additions, and radiocarbon dates indicate. As we documented through excavation, the reuse of HCT monuments occurred sometimes millennia after construction and disuse. This circumstance extended the HCT timeline in Hadramawt (pink dotted line in Fig 2). Elsewhere in Oman, Williams and Gregoricka [40] document re-visits and deposits subsequent to SCAB construction. Our SCAB sample (n = 23) is undated, but we find some that share attributes of Neolithic platforms, such as a standing stone and East-facing façade (e.g., D106-002); others are only some meters from Neolithic platforms (e.g., D027-001, D027-003). (S2 Table, S3 Table)

## Environmental data analysis and machine learning

To determine the factors influencing monument placement, we performed bootstrap aggregating of classification trees and regularized multinomial logistic regression. We extracted cell values from 12 geospatial raster datasets at the 371 monument locations and from a random selection of 500 true absence locations within the viewshed. The environmental raster layers are listed in Table 1 and hosted in the PANGAEA data repository [78]. Topographic variables were derived from the elevation layer using the Geomorphometry and Gradient Metrics toolbox [64] in ArcGIS [79].

We calculated path distances between monuments and each of human routes and springs. Path distances (anisotropic) accounted for cost distances in travelling upslope versus downslope. The cost raster generated in GIS, and used to calculate path distances, used a symmetric inverse linear as the vertical factor setting, which sets an exponentially increasing cost with slope steepness. We identified springs through place name lists and intensive field coverage targeting

**Table 1. The independent environmental variables used in the analysis.**

| Independent variable | Description of variable |
| --- | --- |
| Eastness | Aspect decomposed to two directions, with highest values on slopes facing east and lowest values on slopes facing west. |
| Elevation | An elevation raster layer derived from ASTER Global Digital Elevation Map V2. The original data has been reprojected and resampled using bilinear interpolation. |
| Fog density | A multiband (R, G, B, NIR) raster layer of spatial variability in monsoon fog density calculated on a per cell basis as the mean of the fog reflectance values of 119 Landsat 5 TM scenes, 17 Landsat 7 ETM+ scenes and 121 Landsat 8 OLI TIRS scenes. |
| Normalized Difference Vegetation Index (NDVI) Maximum Greenest Pixel | A max-NDVI raster layer where pixel value is the maximum greenest pixel of all NDVI layers derived from Landsat 8. |
| Northness | Aspect decomposed to two directions, with highest values on slopes facing north and lowest values on slopes facing south. |
| Path distance to roads | Path distances were calculated from a cost raster in ArcGIS. A cost raster identifies the cost of travelling through each cell. In our cost raster, the cost of travel increased exponentially with slope steepness. |
| Path distance to springs | |
| Slope [75] | A raster layer of slope, in degrees, derived from the elevation layer using the average maximum technique. |
| Terrain Ruggedness Index (TRI) [76] | A TRI raster layer derived from the elevation layer, based on the sum change in elevation between a central cell and its eight neighboring cells. TRI is a terrain roughness metric. |
| Topographic Position Index (TPI) (R = 200, 1000, 5000 m) (Guisan et al., 1999) [77] | TPI raster layers with radii (R) of 200 m, 1 km and 5 km, derived from the elevation layer. TPI measures the difference between a central cell elevation and the average elevation around it within a predetermined radius (R). At small scales it is a measure of terrain roughness and at large scales it describes slope position and landform types. |

spring sites [80]. The distributions of travertines attest to former springs, and spring output today is lower than in the past. Springs that persist today in the Nejd were likely also flowing in the past and were therefore particularly reliable water sources around which we expect human activity to be high.

Vegetation was measured by calculating Normalized Difference Vegetation Index (NDVI) from all available imagery in the Landsat 8 surface reflectance imagery collection in Google Earth Engine after removal of very cloudy images (>30%) and cloud-masking. A map layer of the maximum NDVI values was used for analysis rather than NDVI because values are generally very low and highly variable in arid environments.

Fog density quantifies the average distribution of monsoon fog in the study area. The layer was calculated on a per cell basis as the mean of the near-infrared fog reflectance values of 119 Landsat 5 TM scenes, 17 Landsat 7 ETM+ scenes and 121 Landsat 8 OLI TIRS scenes [81].

Maximum NDVI and fog density correlate with a major bio-climatic zonation in Dhofar, which persists due to the interplay between topography and monsoonal advection fog [81,82]. The plateau and the escarpment would always have been moister than the interior and form a natural axis against which to measure regional distributions in the arid interior. We also include eight topographic variables in our models because topographic factors such as aspect, visibility and/or accessibility may have influenced more localized monument placement [59,67]. We evaluated variables similar to those tested elsewhere, including in spatial modelling of ancient funerary monuments in eastern Sudan [2].

**Bootstrap aggregating.** Bootstrap aggregating (bagging) is a machine learning method that averages the results of multiple classification trees, each trained on different random (bootstrap) samples. Combining the results of multiple models helps to improve accuracy and reduce overfitting, compared to using a single tree. Combining the results of multiple iterations of models is common and helps improve accuracy by balancing out each model's strengths and weaknesses, leading to better, more reliable predictions. Classification trees split data into smaller subgroups through recursive partitioning, based on true or false answers about the values of predictors. Each split is based on a single variable, and a rule that maximizes the number of similar values (class purity) within each of the two resulting subgroups [83]. For example, a classification tree might split animals by weight, classifying those heavier than 10 kg as mammals and those lighter as birds. Classification trees are well-suited to geospatial datasets where nonlinear responses and predictors may interact in unknown ways [84]. In bootstrap aggregating, classification trees are grown until all the observations have been classified, unlike single trees where a complexity parameter limits tree growth. Bootstrap aggregating was performed in the R 'rpart' package [85,86] which implements methodologies of Breiman *et al.* [83]. We chose to generate a high number of trees (500) as the results of single trees differed greatly. Model accuracy is assessed using the out-of-bag misclassification error. Variable importance was calculated as the sum of reductions in mean-squared error each time a variable was used as a primary or surrogate split and summed across all trees. For each variable, partial dependence plots were generated to visualize the relationships between the environmental variables and the monument locations [87]. (S1 Fig)

### Environmental data analysis and regularized multinomial logistic regression

To examine the power of the environmental variables for predicting monument types and to understand the direction and magnitude of these relationships, we performed regularized multinomial logistic regression. This method was preferred over machine learning techniques such as bootstrap aggregating, as it is more robust to the unequal numbers of observations of different monument types. It also incorporates variable selection and provides highly interpretable results. Regularized multinomial logistic regression was performed in the 'glmnet' and 'glmnetUtils' packages in R [85,88]. A range of regularization techniques can be applied, from ridge regression [89], to elastic net [90], to Least Absolute Shrinkage and Selection Operator (LASSO) [91]. Parameter α determines the type of regularization. When α = 0, ridge regression shrinks the coefficients of correlated variables toward each other but retains them all. When α = 1, LASSO shrinks coefficients, producing a sparse model by filtering out less important variables. When α => 0 or < 1, elastic net regression balances

ridge and LASSO, allowing both variable selection and handling of collinear predictors. Both LASSO and elastic net can be used implicitly for variable selection [92]. A second parameter, λ, controls the overall penalty strength. As λ increases, more coefficients shrink towards zero, further refining the model.

All variables were standardized to have a mean of zero and a variance of one before analysis to ensure comparable model coefficients. We performed K-fold cross validation in the R 'glmnetUtils' package to determine optimal values for α and λ, using deviance as the performance metric. The model with the lowest multinomial deviance identified $\alpha = 0.4$ (elastic net regression) (S2 Fig). We report coefficients from the model with the largest λ within one standard-error of the minimum λ value ($\lambda = 0.049$, log $\lambda = -3.023$) (S3 Fig). This follows the one-standard-error rule [88], the default in glmnet, which accounts for estimation uncertainty in risk curves.

We also used regularized multiple linear regression to examine how environmental factors influence total stone volume, providing insight into the relationship between environmental conditions and labor input. If monument builders selected locations to communicate something about their significance, we would expect greater labor investment in environmentally important areas.

As before, we log-transformed stone volume for normality and standardized all variables. K-fold cross-validation identified $\alpha = 0.1$ (elastic net regression) based on the model with the lowest mean-squared error (S4 Fig). We compare results from both the minimum λ and one-standard error λ models (S5 Fig). Since the one-standard error model with no variables showed similar accuracy and error to the minimum λ model, the latter's results should be interpreted with caution. We repeated this analysis using only data of HCTs and triliths, the two most common monument types.

## Collocation analysis

Using our regional chronological sequence of monument types, we also tested spatiotemporal relationships between monument types. To communicate something about historical place and builders' attachments to it, monument builders may have decided to build where prior monuments existed. Clustering would suggest that builders were reinforcing or adjusting prior perceptions of landscape through monument placement. We used collocation analysis to test whether the locations of preexisting monument types influenced the placement of subsequently constructed monuments [93]. This technique determines the degree to which a subsequent monument type is collocated with or isolated from the preceding monument type, relative to other monument types, in the neighboring 20 monument locations.

In addition, we tested for spatiotemporal relationships between the monument types. We calculated the spatial mean center and standard distance of the monument types to observe general distributional trends of the different monument types. The mean center is the geographic center, calculated by averaging the monument coordinates. Standard distance measures the degree to which the monuments are concentrated or dispersed around the mean center. We used a distance of one standard deviation from the mean. We can visually compare these measures to observe geographic overlap or dispersion of monument types.

## Technological and labor data analysis

We tested whether the technological and labor demands of monument construction followed a trend with aridification. We used the total stone volume of a monument as an indicator of labor input [66,94,95]. Using ANOVA with Tukey post-hoc tests, we compared total stone volumes across different monument types - platforms, SCABS, HCT, triliths, and boat graves -which represent distinct chronological periods.

Since labor can be provided simultaneously by many or sequentially by few, we test whether earlier monuments also have the largest stones, a proxy for simultaneous labor. If monuments with larger total stone volumes also include the largest stones, it would support prior research suggesting that large groups convened episodically during the wetter Early Holocene to simultaneously invest labor for monument building [29,53].

We further used ANOVA with Tukey post-hoc tests, to compare total stone volumes between monuments that included one or more large stones and those that did not, repeating the test within each monument type.

We also compared total stone volumes between two categories of construction-- -*episodic* and *accretive*--for insight into the relationship between labor investment and collective action. We already understand the temporal sequence: episodic construction pre-dates accretive construction. Neolithic platforms had to be assembled in one episode to stand. Bronze Age HCT served as tombs, and their architectural and stratigraphic characteristics suggest they were also constructed in one episode. SCABs could have been accretive, but the documented examples are all complete, hinting that they never were left unfinished as might happen in accretive, sequential, low investments of labor. Accretive triliths were built with elements that could be assembled by few people in short time but that could be replicated thereafter by newcomers; indeed, there are examples of triliths with a lone element and examples with dozens. boat graves are similar.

To test whether large, simultaneous groups built early episodic monuments, we also tested the association between size of the largest stone and technological requirements, using a chi-squared test of independence.

## Results

### Environmental factors influencing monument placement

Monuments occur in a region now hyper arid, characterized by sparse water availability and uneven graze. Our survey yielded areas with high monument concentrations—around the oasis of Mudhai, the Wadi Haluf-Hanun corridor, and two tributaries of Wadi Dhahabun. (Fig 3) Analytical results show how local environmental factors explain the positioning of monuments in the landscape and whether cultural factors like routes and extant monuments were significant.

Supported by machine learning, our analysis revealed a strong spatial patterning in the environmental characteristics of 371 monument locations. Results from bootstrap aggregating show that (Fig 4a) proximity to springs was important, whilst a partial dependence plot shows that the probability of monument occurrence decreases substantially beyond 5 km from springs (Fig 4b). Most monuments were also placed in low topographic positions, within wadis or depressions (Fig 4b). The probability of any monument occurrence decreases with increasing distance from routes (Fig 4b), with most monuments located within a few hundred meters of routes. Furthermore, the probability of monument occurrence increases sharply with vegetation (max-NDVI) in arid areas (S1 Fig).

Environmental differences are apparent across the locations of different monument types characteristic of different time periods. A multinomial logistic regression preserved ten non-zero variables to predict monument type with coefficients (Fig 5) for each environmental variable for each monument type. Coefficient path plots show changes in the coefficient values with decreasing λ (S6 Fig).

Our results in Figure 5 show that platform monuments tend to occur at lower elevations, close to springs and in low topographic positions in localized context (r=200 m), such as within small wadis and depressions. However, platforms do not show an association with any particular topographic position at larger geographic scales. HCT monuments are more frequent in areas with high fog densities, with low NDVI values, on north-facing slopes. They tend to occur close to routes but farther from springs. Moreover, HCTs occupy steep slopes in high topographic positions of large landforms, such as in plateau areas and on the shoulders of large wadis. Trilith monuments occur in low topographic positions within wadis and depressions. They are associated with areas that have low fog densities and low vegetation cover, on north-facing slopes close to springs and routes. We find that boat grave monuments are associated with areas at higher elevations, with high fog densities and high vegetation cover. They tend to occupy south-facing slopes in high topographic positions of small landforms.

Eleven of the twelve environmental variables were preserved in the minimum lambda model to assess whether labor input responded to environmental factors. The variables with the highest coefficients entered the model early and have the greatest predictive power (S7 Fig). Notably, labor input (total stone volume) increased with fog density and topographic position at large scales, but decreased with maximum NDVI (S8 Fig). This can be explained by social responses to population densities, such that people in the mountains had frequent enough contact that monuments were not needed as social

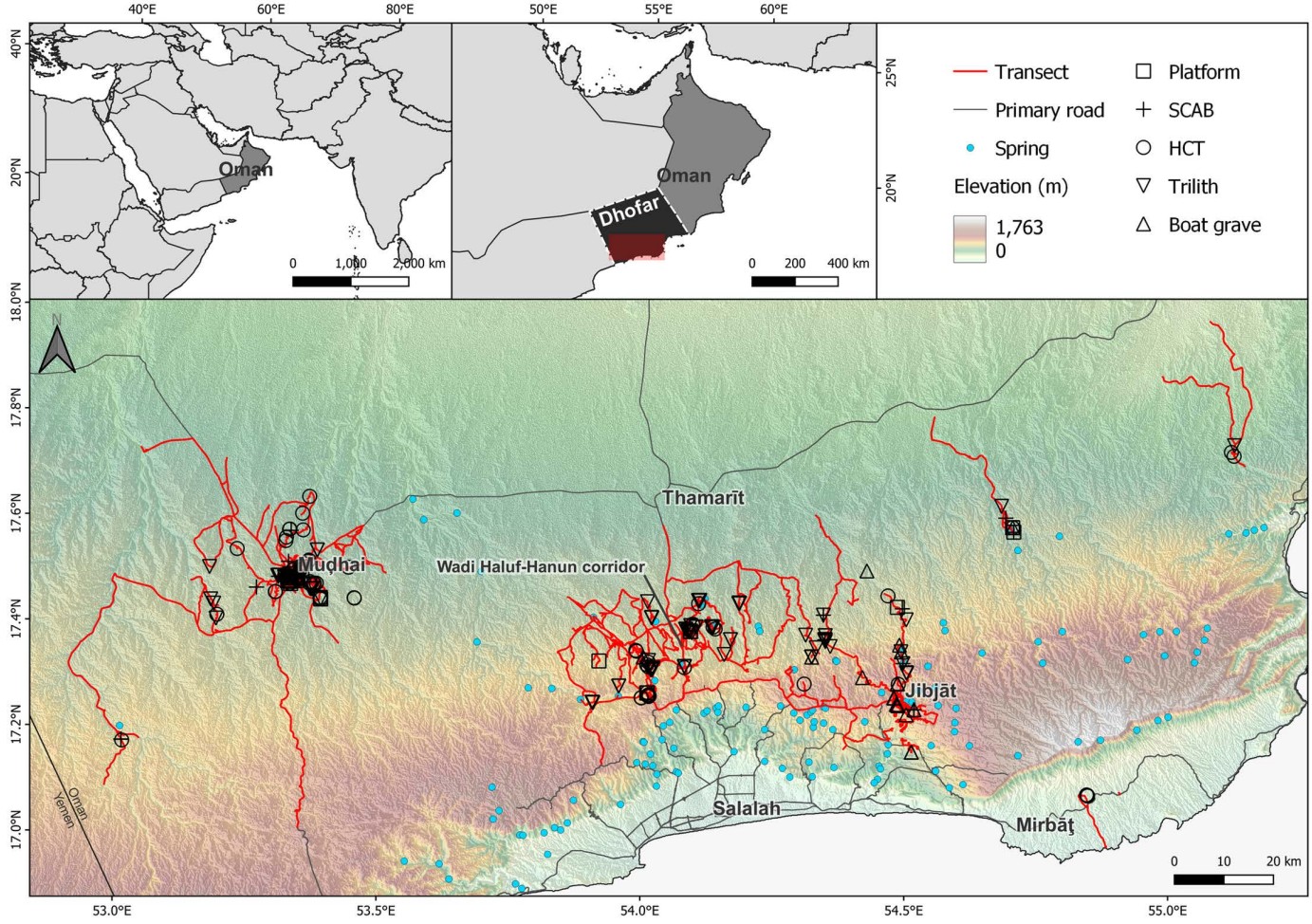

**Fig 3. The locations of monuments, survey transects, primary roads and springs in Dhofar.**

touchstones. We also analyzed whether labor input responded to environmental factors within the most common monument types; HCTs (n = 100) and triliths (n = 138). No variables were preserved in the 'one-standard error' rule models, but in the minimum lambda models, total stone volumes for HCTs increased with fog density (coef = 0.0798), whilst for triliths, total stone volumes decreased with increasing slope steepness (coef = -0.0677). These results are also consistent with an interpretation that HCT (episodic) lay within a zone where labor could be convened, closer to the vegetated mountains, while the accretive triliths decline in size where labor expenditures rise in carrying stones up and down steep inclines.

## Spatiotemporal relationships between monuments

Across Dhofar, we documented monument clusters in areas close to important water sources today; the twin springs at Mudhai oasis and the springs at Ayūn, Halūf, and-Hanūn. A visual comparison of the mean center and standard distance (1 SD) of the monument types shows that known boat graves are localized in eastern Dhofar, whilst the other monument types have overlapping, broad distributions (Fig 6). HCT and trilith monuments are more common than the other monument types and occur across the greatest geographical range. Conversely, platform and SCAB monuments occur in low numbers, although the latter show a highly dispersed distribution.

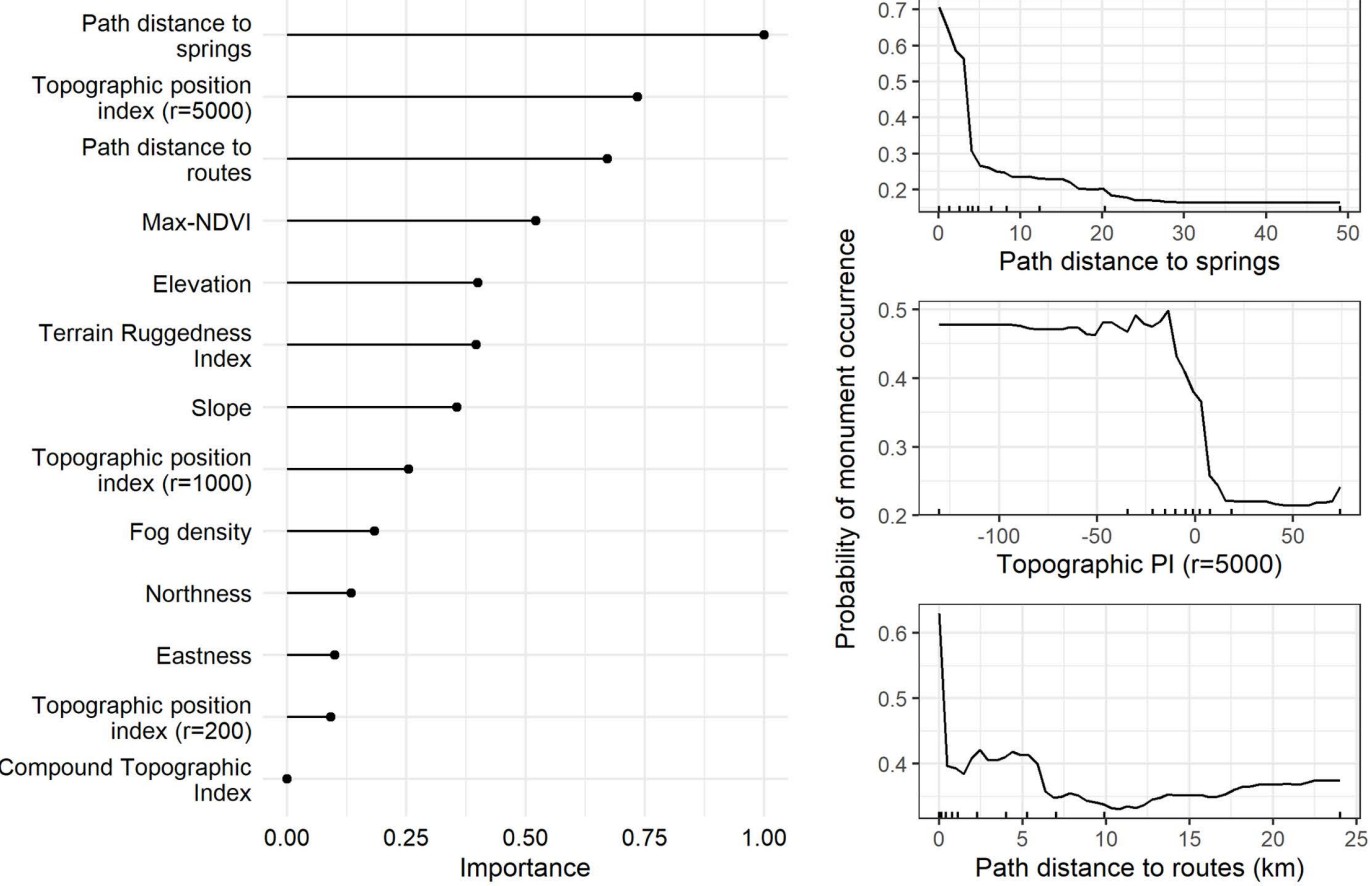

**Fig 4. a. Variable importance from the bootstrap aggregating (Left); 4b. Partial dependence plots, with deciles for data points, show the relationship between environmental variables and the predicted probability of monument occurrence (Right).**

Did pre-existing places marked by monuments increase the likelihood that later people would build there? Because people respond to cultural places, we expected that monument builders would respond to prior locations and influence future perceptions. To our surprise, the results of the collocation analysis in Table 2 show that very few monuments of the successively occurring monument type are collocated with the chronologically preceding monument type. Indeed, when comparing the two most common monument types, we see that a large proportion of trilith monuments are significantly isolated from HCT monuments (HCT are on ridgelines, triliths on low terraces).

### Technological and labor requirements for monument construction

We considered the overall labor contributions to different monuments over time and what, if any, of it, had to be simultaneous labor (episodic construction). Within each monument type (Fig 7, right) and within each of two categories of technological requirement (Fig 7, left), we found no significant difference in total stone volume depending on the size of the largest stone (< 0.75 m or > 0.75 m, paired blue and yellow distributions).

However, if all monuments are considered together (Fig 7, center), those with one or more large stones (> 0.75 m) had significantly higher ($F(1,289)$ = 5.383, $p$ = 0.021) total stone volumes than those with only small stones (< 0.75 m). A chi-squared test of independence determined a significant association between the size of the largest stone and the

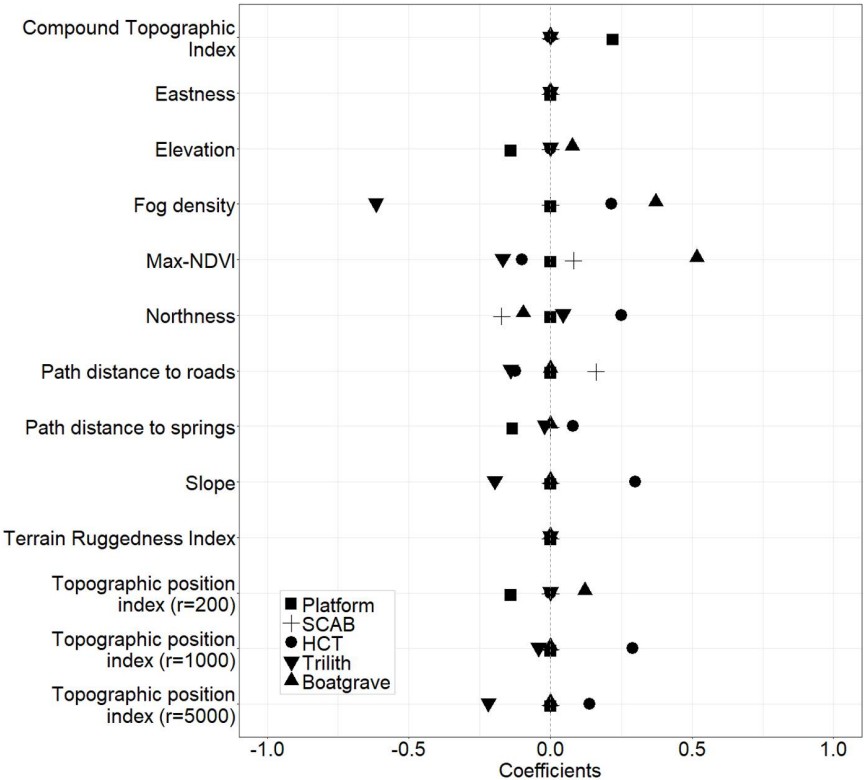

**Fig 5. The regularized multinomial logistic regression model coefficients show the relationship between the environmental variables and the monument types.** Standard errors cannot be accurately estimated in regularized logistic regression. A positive coefficient indicates that the monument type is associated with higher values of that variable, whilst a negative coefficient shows that the monument type is associated with lower values of that variable.

technological requirements ($X2$ (1, N = 298) = 36.994, p < 0.001). Single-episode constructions are significantly correlated with larger volumes (Fig 7, left), which is consistent with a large assembly of people. Conversely, accretive constructions are correlated with smaller monument volume, which is consistent with small groups and sequential labor. The greater labor invested in large platforms involved simultaneous labor—more adults to lift the large stones.

We found chronological patterns. Our survey found fewer platforms than other monument types, a fraction of the HCT or triliths counted, despite the longer timespan during which platforms were built. Within our sample, roughly one platform per 100 years contrasts with one HCT every 12 years and one trilith (full line of elements) every four (the frequency of triliths increases if each plinth is counted as an accretive element). Furthermore, an ANOVA with Tukey post-hoc test showed that platforms have significantly higher ($F(4,291)$ = 6.034, $p$ = 0.00011) total stone volumes than the later monument types (Fig 7, right) and thereby represent the greatest cumulative labor investment.

With HCT, a subsequent pulse of monument construction began around 5200 cal BP. Because HCT would not hold up if built in accretive phases, their construction also indicates single-episode construction (which makes sense when burying a corpse in the heat). There are exceptions [20]. Nonetheless, HCT have significantly smaller stone volumes than platforms.

The next chronological-cultural pulse—accretive trilith monuments--follows increasingly arid conditions beginning around 2700 cal BP [43,44]. Monuments with accretive construction (triliths, boat graves) have significantly lower ($F(1,294)$ = 9.776, $p$ = 0.00194) total stone volumes, compared to those with episodic construction. Fig 7 shows that accretive monuments, such as the numerous triliths, have few large stones (> 0.75 m).

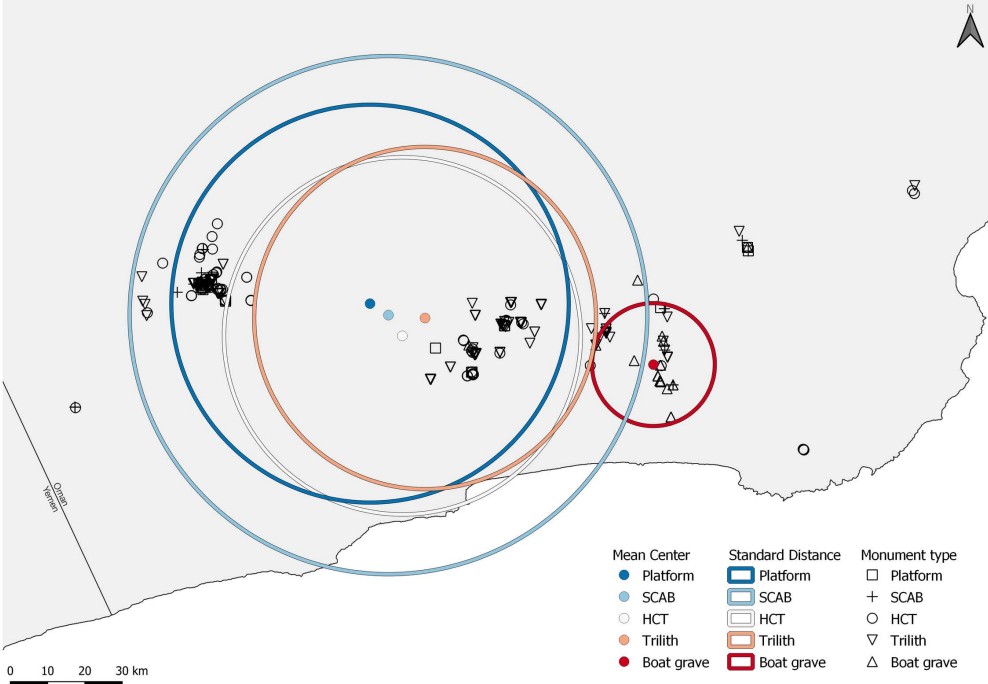

**Fig 6. The mean center and standard distance (1 SD) of the monument types.** The mean center is shown as a labelled colored point with the corresponding circle showing standard distance. Platforms (green), Scabs (orange), HCTs (blue), Triliths (purple) and Boat graves (red).

**Table 2. The results of the collocation analysis.**

| Chronologically preceding type | Chronologically successive type | Proportion of monuments significantly collocated | Proportion of monuments significantly isolated |
|---|---|---|---|
| Platforms (7500–6000 cal BP) | SCABS (7500–5000 cal BP) | 4/23 | 0/23 |
| SCABS (7500–5000 cal BP) | HCT (5200–4000 cal BP) | 8/135 | 0/135 |
| HCT (5200–4000 cal BP) | Trilith (2300–1700 cal BP) | 0/165 | 100/165 |
| Trilith (2300–1600 cal BP) | Boat graves (1100–750 cal BP) | 1/22 | 12/22 |

Accretive construction continued with a new syntax as boat graves after 1100 cal BP. Although each element (a grave) was assembled as one episode, our measured volumes are the total accretive clusters of graves, accumulated as successive burial events. While different in geographical range, in total stone volume these clusters do not significantly differ from the triliths that preceded them.

## Discussion

Over previous decades, archaeologists have studied desert monuments as singular types and classified their forms and uses—*mustatils*, graves, tombs, alignments, triliths, tumuli, megaliths, burials, sacrifices, markers, ritual theaters—without engaging a holistic perspective. Monuments had culturally specific meanings [53,96]. Seeing all these features as monuments focuses on their common potential as proxies for patterned human behavior. By emphasizing and quantifying common features, we arrive at an empirical model and synthetic understanding of dedicated pastoralists' use of monument technology for social and community resilience to climate change.

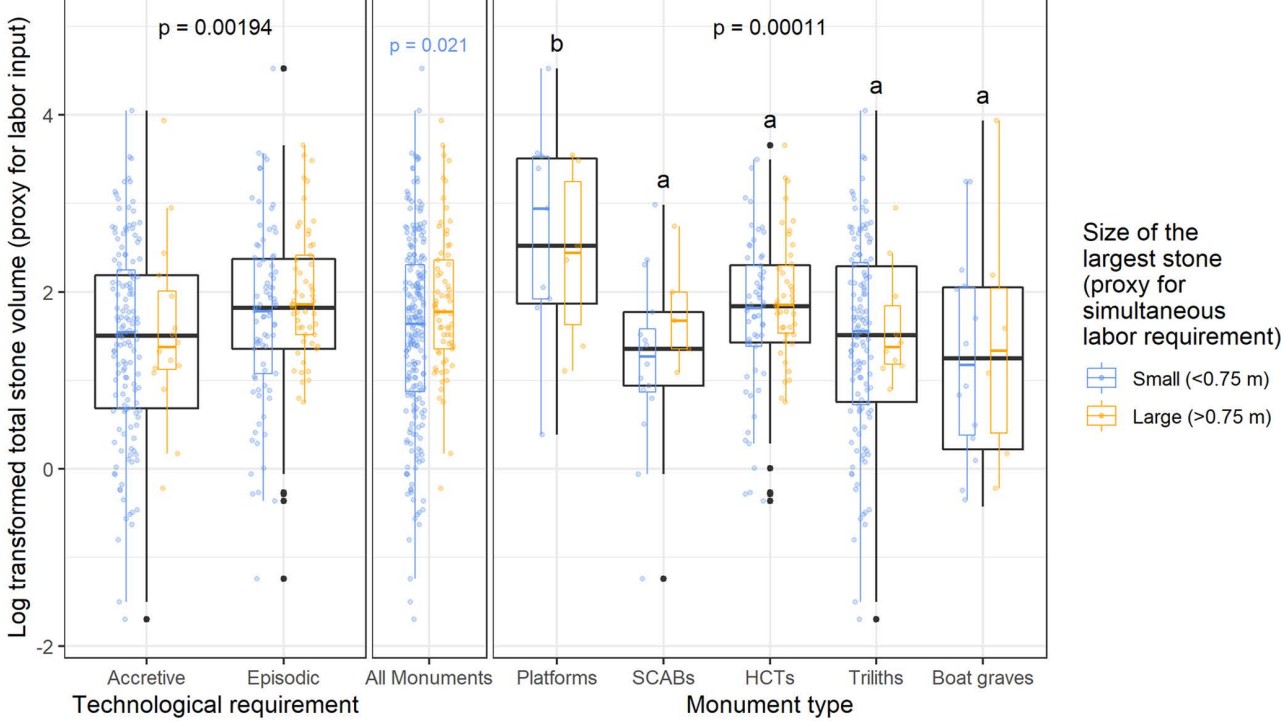

**Fig 7. Boxplots comparing total stone volumes across several monument groupings.** Groupings include the five monument types, the two categories of technological requirement, and whether or not one or more large stones (>0.75 m) were present. We also compare the latter for all monuments, within monument types, and the two categories of technological requirement. The letters show the results of Tukey HSD post hoc pairwise comparisons following ANOVA tests. Monument types with letters in common indicate no statistically significant differences, while different letters indicate statistically significant differences.

Social resilience emerges from coping, adaptive, and transformative capacities to "tolerate, absorb or adjust to environmental and social threats" [97, p.8]. These are capacities all linked at the community level through network densities that "generate and produce assets that advance shared objectives" [98, p.3]. We consider monuments as assets that facilitated social belonging and were technically adjustable to sustain social cohesion for desert pastoralists who did not abandon herding with environmental changes. Such changes included reduced graze, drying springs, changes in frequency and distributions of seeps, and in some eras, amelioration of rangeland.

In contrast to people occupying the vegetated plateau and escarpment, populations in the arid zone relied on monument construction and subsequent visits for social contacts that defined their socioeconomic and cultural networks. We attribute monuments' distribution to their roles in communicating socio-cultural messages to people often absent. Monuments communicate a culturally specific, shared meaning from builders to contemporaries and facilitate exchange and social relationships [99]. Builders may also have intended to signal subsequent generations or to reify builders' attachments to landscapes and places already established by prior generations.

The placement of monuments could communicate environmental information like grazing or routes or could be reactive to and built alongside an established route. Monument placement close to water and streambeds, where vegetation is highest, reflects that these are the places desert people frequent. These low-lying areas offer graze and browse, shade under trees, rockshelters, and travel routes of least effort.

With consistently higher biomass in the escarpment through time, populations there were probably denser than in the arid interior. Dhofar's mountain people could rely on direct contact to build social networks for resilient communities, rather

than rely on the technologies of communication involving monuments with culturally specific construction and mnemonics[42]. Moreover, cloud forests and tall grasses on the escarpment limited visibility, thereby limiting the utility of a monument. (Ancient houses, corrals, and hearths are visible, so preservation and sampling do not prevent archaeologists from detecting monuments).

Large differences exist in the numbers of examples we recorded for each monument type. Prior experience in a comparable landscape (Hadramawt, southern Yemen) demonstrated that these numbers are not significantly explained by preservation conditions of different landforms. Moreover, platforms are significantly associated with lower elevations and HCT in higher elevations [58], both captured in our viewshed boundary in this research so that our monument counts would be relatively unaffected by terrain differences. Therefore other factors—such as period, cultural conventions, available labor—account for differences in the numbers of monuments of each type.

Results of collocation and environmental analyses suggest that historic place attachments alone cannot explain spatial distributions of different monument types. A shift from earlier platforms close to springs to HCT built further from springs coincides with major aridification. Perhaps desert people no longer clustered as close to reliable water (and there is no evidence they adopted oasis farming in Dhofar). Instead, the shift reflects adaptive resilience through major subsistence change during an era when goats replaced cattle in the arid interior [100, p. 228–9]. Goats can go longer without water and browse a wider range of forage [101]. HCT are also located on elevated sites compatible with the steep terrain of goats. The northern face of a slope would have less vegetation, given that prevailing moisture comes as fog from the south, so HCT would have higher prominence on bare terrain, perhaps consonant with the territorial signaling archaeologists have suggested as their function [8,102]. HCTs and triliths occurring across a broader geographical range are consistent with populations unable to support clustered herds on the sparse graze available near springs and adapting through dispersal.

This adjustment in pastoralism (while eschewing farming) raises the role of monuments in the adaptive social resilience of a dispersed, thinned, Middle Holocene Dhofari population responding to climate change. Harrower's analysis of water management in Wadi Sana, Hadramawt, noted the contemporaneity of HCTs and water management in a drying landscape, speculating that "death and ancestry-infused landscape markers maintained deep social importance" [102, p. 505]. To other archaeologists also, the appearance of Bronze Age tombs suggests that resilient people now forged social identities and mediated access to resources through kinship with memorialized dead [8,34]. Our analysis scrutinizes and quantifies these purported relationships, explaining why monuments of different eras do not appear to cluster as historic attachment to place.

People's choice to build different styles in different places signals or reflects a changing social landscape in different eras. Collocation analysis and the significantly different local environments of different monument types support the chronological framework developed from a dated sample of monuments. While triliths extend beyond our study area and may mark the once-expanded range of a particular ethnic group [74,103, p.73-4], the distribution of boat graves is restricted to eastern Dhofar, with possible parallels elsewhere [39]. Anecdotally, our team noted very few instances in which early monuments were quarried in antiquity to erect later monuments. If people did respond to prior monument placement at all, it was to place some later monuments apart from their antecedents, not to rebuild close by. We suggest that the technology of monuments communicating culturally specific social identities was a persistent feature of pastoral groups, even as the messages that monuments communicated varied through time and space.

Changes in the volume and size of stones used in construction point to variable labor investments. Containing larger stones, Neolithic platforms are the largest and earliest single-episode monuments. To assemble such a platform requires balancing uprights with interior chock stones until all are held in place by a rapidly accumulated rubble fill. The construction sequence suggests that simultaneous cooperation among multiple herding households provided labor for these significantly larger monuments. At minimum, the group size could be several families, with seven strong adults to lift the largest stones. At one Neolithic platform in Yemen, rare preservation conditions indicate gatherings of up to 5000 people [29,53,71]. Ethnographic case studies document episodic monument construction events that aim to showcase an

especially large amassing of labor [104–106]. Construction of platforms commemorating such gatherings ended around 6200 cal BP, as the Southwest monsoon influence began to wane and with it, vegetation, animal life, and human population densities in the Arabian interior [26,107]. Perhaps it became more difficult to gather and feed a large, simultaneous labor force.

Thereafter, building HCT required shifting less stone, which we suggest reflects smaller groups in thinned, still mobile populations responding to aridity. The new practice of burying the dead in monuments--now tombs--reflects a resilient strategy emphasizing people's social connections through genealogy and the transfer of more complex skills in construction. By memorializing the dead, people created touchstones for living lineages, even as people were less likely to gather in large social groups and clung to regions of fog density and seasonal graze [8,24,55,102]. This threshold in social constitution archaeologically anchors the beginnings of genealogical time-depths, consequential to this day across arid regions of Arabia, Africa, and Central Asia. Many label this social framework, "tribes" [108–111], but ethnographic examples are, at best, rudimentary analogs for social configurations of the distant archaeological past.

Higher frequency and smaller stone volume of accretive triliths with few heavy stones is consistent with monuments built incrementally by smaller, dispersed groups from a reduced population and persistent in an era of hyper-aridity. In the deserts, scattered people signaled their social affinities with culturally distinct syntax and built themselves into communities and places, even perhaps in the absence of actual people. Triliths (2300–1700 cal BP) could be built as accumulations of distinct elements by successive visitors, a few at a time. The high variance in trilith volumes hints at this; even as the iconic trilith form was relatively consistent over a large region [112,113], some triliths had one element while others had 40. Triliths appear in extremely arid zones after the introduction of camels, which extended human range and herder intervals between water sources.

Accretive triliths and boat graves appear in the wake of important climate changes. Boat graves appear after 1150 cal BP when climate proxies in Southern Oman indicate increased monsoon moisture after a particularly arid period. Unlike previous episodes of monument construction (HCT, triliths), boat graves coincide with a return to slightly moister conditions (Fig 2). Some boat graves appeared in the context of abandoned settlements, where mobile builders minimized labor costs by quarrying nearby buildings [39,114]. People could settle and farm, and we suggest a resilient commitment to mobile pastoralism—for its social and ideological traditions as much as for its ecological inheritance.

These changes in monuments coincided with climate thresholds in Holocene Arabia, yet change was not always adjustment to aridification. Instead, we see changes in monument location, construction, and requisite labor parties as shifts in the criteria for participation in mobile, monument-linked communities adjusting to various environmental and social stressors. We suggest that such communities rallied at population density thresholds where people lacked habitual contact, whether through aridification (HCT, triliths) or a wetter era that relaxed water constraints on mobility and herds (boat graves).

Southern Arabia is a significant case that informs our contemporary challenges by showcasing the adaptive social resilience of people committed to specific economic strategies, like pastoralism or small-scale farming, in the face of climate and environmental dynamics. Across the African Sahel, monument construction appears among pastoralists from the earliest Neolithic [115] and was manifest, with adjustments, through time [21,116]. In the American Southwest, pueblo development has been linked to drier and more variable conditions [117,118]. Much prior focus has been on the establishment of Arabian oasis sedentism with changes in economy and mobility [119]. We offer a view of the behaviors that sustained desert pastoralism.

Our analysis suggests socioeconomic resiliency to a dynamic environment. All mobile pastoralists need communities to exchange information, animals and other goods, reproductive partners, and commitments. These circumstances forced repeated adjustment in the technologies and institutions of group identity even as pastoralism remained an economic mainstay. We accept prior claims that Bronze Age tomb-building reflects ancestry as a foundation of social identity and resource access [8], and we accept the appearance of tombs and oases (beyond Dhofar) correlated with climate change [24,26]. By linking all monuments, our analysis newly demonstrates that this was a resilient social adaptation from earlier,

large, social gatherings of contemporaries living in a more abundant environment. Furthermore, the appearance of Arabian genealogical organization as adaptive social resilience belongs to a larger pattern: over time, ever-smaller groups fragmenting into culturally distinct regions began to build accretive monuments to maintain collective social identities among persistent pastoralists who seldom, if ever, coalesced.

## Conclusions

Derived from the metrics of labor scheduling and location, and testable in new data sets, our empirical model theorizes eras of monument building as the exercises of social capacities to absorb risk in the dynamic climates and environments of the arid sub-tropics of Arabia. By decentering analysis from a single era or type of monument, we show how monument classes are proxies for social behavior of mobile, persistent pastoralists. Our model highlights a reliance on monuments as touchstones of social belonging and as flexible technologies of social resilience in a changing world. While built on archaeological observations of monuments in Arabia, our model may be applicable and adaptable to assess social resilience in other regions, such as Saharan, Mongolian, or the high Andes.

## Supporting information

**S1 Table. Codes and Metrics of 371 monuments in Dhofar used in this analysis.**
(XLSX)

**S2 Table. Description of monument types with age ranges of archaeological periods.**
(XLSX)

**S3 Table. Samples, contexts, and radiocarbon determinations shaping the chronology of monuments in Dhofar.**
For comparison with published Bayesian analysis of radiocarbon determinations [38], see horizontal bars in Fig 2.
(XLSX)

**S1 Fig. Partial dependence plots showing the relationship between each environmental variable and the predicted probability of monument occurrence.**
(PDF)

**S2 Fig. Multinomial deviance across values of log λ from 11 models with different α parameter settings.** We used the model with α = 0.4 (elastic net regression), which had the lowest multinomial deviance.
(PDF)

**S3 Fig. Multinomial deviance across values of log λ with α = 0.4 (elastic net regression).** We used the 'one-standard error' rule, to select the model with λ = 0.049 (log λ = -3.023).
(PDF)

**S4 Fig. Mean-squared error across values of log λ from 11 models with different α parameter settings.** We used the model with α = 0.1 (elastic net regression), which had the lowest mean-squared error.
(PDF)

**S5 Fig. Mean-squared error across values of log λ with α = 0.1 (elastic net regression).** We selected the model with the minimum λ = 0.129 (log λ = -2.047).
(PDF)

**S6 Fig. The elastic net regression coefficient paths for each monument type.** As λ decreases, the maximum permissible value of the L1 norm increases and more coefficients enter the model. Variables enter the model based on their true linear regression coefficient, and therefore variables that enter the model early have a higher predictive power, compared

to those that enter the model later. When a new variable enters the model, it affects the slope of the coefficient paths of the other predictors, depending on the magnitude of collinearity.
(PDF)

**S7 Fig. The elastic net regression coefficient paths for largest stone volume. Penalized coefficients are colored grey.** The regularized multiple linear regression assessed whether labor input responded to environmental factors; however, using the 'one-standard error' rule, no variables were preserved in the model. Eleven of the twelve environmental variables were preserved in the minimum lambda model. The variables with the highest coefficients entered the model early and have the greatest predictive power. Notably, labor input (total stone volume) increased with fog density and topographic position at large scales but decreased with maximum NDVI. Like the results of bootstrap aggregating, the minimum lambda result suggests a threshold in population dispersal at which humans engage monuments for messaging-while-absent. At the higher population densities supported in the vegetated mountains, people had sufficient direct contact that monuments were not needed as social touchstones. Fog density is high in the near Nejd backslope of the plateau, where mobile families would readily find seasonal grazing and yet be infrequently in contact, using monuments to communicate.
(PDF)

**S8 Fig. The regularized multiple linear regression model coefficients show the relationship between total stone volumes of monuments and the environmental variables.** A positive coefficient indicates that stone volume increases with that variable, whilst a negative coefficient shows an inverse relationship between stone volume and that variable. We also analyzed whether labor input responded to environmental factors within the most common monument types; HCTs (n = 100) and Triliths (n = 138). No variables were preserved in the 'one-standard error' rule models, but in the minimum lambda models, total stone volumes for HCTs increased with fog density (coef = 0.0798), whilst for Triliths, total stone volumes decreased with increasing slope steepness (coef = -0.0677). These results are also consistent with bootstrap aggregating and with an interpretation that HCT (episodic) lay within a zone where labor could be convened, (closer to the vegetated mountains), while the accretive triliths decline in size where labor expenditures rise in carrying stones up and down steep inclines.
(PDF)

## Acknowledgments

The authors are grateful to the Ministry of Heritage and Tourism, Sultanate of Oman; The Ohio State University; General Organization of Antiquities and Museums, Republic of Yemen; the Office of the Advisor to HM The Sultan for a reconnaissance visit; and especially to the teams of colleagues, students, and locals who helped accomplish the work. Thanks also to Shane Scaggs, who completed Fig 2.

## Author contributions

**Conceptualization:** Joy McCorriston.

**Data curation:** Joy McCorriston, Lawrence Ball, Michael J. Harrower, Matthew J. Senn, Abigail F. Buffington.

**Formal analysis:** Lawrence Ball, Ian M. Hamilton.

**Funding acquisition:** Joy McCorriston, Ian M. Hamilton.

**Investigation:** Joy McCorriston, Michael J. Harrower, Matthew J. Senn, Tara Steimer-Herbet, Abigail F. Buffington, 'Ali Ahmad Al-Kathiri, 'Ali Musalam Al-Mahri.

**Methodology:** Joy McCorriston, Lawrence Ball, Michael J. Harrower, Ian M. Hamilton, Matthew J. Senn, Tara Steimer-Herbet, Abigail F. Buffington.

**Project administration:** Joy McCorriston, Ian M. Hamilton.

**Resources:** Joy McCorriston, Ian M. Hamilton, 'Ali Ahmad Al-Kathiri, 'Ali Musalam Al-Mahri.

**Supervision:** Joy McCorriston, Ian M. Hamilton.

**Visualization:** Lawrence Ball.

**Writing – original draft:** Joy McCorriston, Lawrence Ball, Sarah J. Ivory.

**Writing – review & editing:** Joy McCorriston, Lawrence Ball, Michael J. Harrower, Ian M. Hamilton, Sarah J. Ivory, Tara Steimer-Herbet, Abigail F. Buffington.

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
