## [Decision Letter · Decision Letter 0]

14 Jan 2025

PONE-D-24-34740South Arabia’s prehistoric monument landscape shows social resilience to climate changePLOS ONE

Dear Dr. McCorriston,

Thank you for submitting your manuscript to PLOS ONE. After careful consideration, we feel that it has merit but does not fully meet PLOS ONE’s publication criteria as it currently stands. Therefore, we invite you to submit a revised version of the manuscript that addresses the points raised during the review process.

We look forward to receiving your revised manuscript.

Kind regards,

Shai Gordin, Ph.D.

Academic Editor

PLOS ONE

Journal Requirements:

3. We noted in your submission details that a portion of your manuscript may have been presented or published elsewhere. [Details of individual tomb excavations and multiple radiocarbon dates were published previously in McCorriston 2023 Persistent Pastoralism (book). The published details serve as foundational information for the broader, collective analysis developed here, which relies on unpublished survey data and original analysis. While excavations of most individual tombs have therefore previously appeared, the prior publication--including vital stratigraphic and construction details--underpins classification of monument types and contexts for radiocarbon dates in this manuscript. Such detail is far too lengthy (book) to include in PLOS ONE but serves as essential transparency for assertions on construction details, classifications, and dating in the PLOS ONE submission]. Please clarify whether this [conference proceeding or publication] was peer-reviewed and formally published. If this work was previously peer-reviewed and published, in the cover letter please provide the reason that this work does not constitute dual publication and should be included in the current manuscript.

5. In the online submission form, you indicated that your data will be submitted to a repository upon acceptance. We strongly recommend all authors deposit their data before acceptance, as the process can be lengthy and hold up publication timelines. Please note that, though access restrictions are acceptable now, your entire minimal dataset will need to be made freely accessible if your manuscript is accepted for publication. This policy applies to all data except where public deposition would breach compliance with the protocol approved by your research ethics board. If you are unable to adhere to our open data policy, please kindly revise your statement to explain your reasoning and we will seek the editor's input on an exemption.

6. We note that there is identifying data in the Supporting Information file <Description of Monument Types.xlsx and S-T3 dates_tableV2_ OMAN MONUMENTS.xlsx>. Due to the inclusion of these potentially identifying data, we have removed this file from your file inventory. Prior to sharing human research participant data, authors should consult with an ethics committee to ensure data are shared in accordance with participant consent and all applicable local laws.

-Location data

Additional guidance on preparing raw data for publication can be found in our Data Policy (https://journals.plos.org/plosone/s/data-availability#loc-human-research-participant-data-and-other-sensitive-data ) and in the following article: http://www.bmj.com/content/340/bmj.c181.long .

Please remove or anonymize all personal information (Name,Date), ensure that the data shared are in accordance with participant consent, and re-upload a fully anonymized data set. Please note that spreadsheet columns with personal information must be removed and not hidden as all hidden columns will appear in the published file.

7. We note that Figures 3 and 6 in your submission contain [map/satellite] images which may be copyrighted. All PLOS content is published under the Creative Commons Attribution License (CC BY 4.0), which means that the manuscript, images, and Supporting Information files will be freely available online, and any third party is permitted to access, download, copy, distribute, and use these materials in any way, even commercially, with proper attribution. For these reasons, we cannot publish previously copyrighted maps or satellite images created using proprietary data, such as Google software (Google Maps, Street View, and Earth). For more information, see our copyright guidelines: http://journals.plos.org/plosone/s/licenses-and-copyright .

1. You may seek permission from the original copyright holder of Figures 3 and 6 to publish the content specifically under the CC BY 4.0 license.

We recommend that you contact the original copyright holder with the Content Permission Form (http://journals.plos.org/plosone/s/file?id=7c09/content-permission-form.pdf ) and the following text:

“I request permission for the open-access journal PLOS ONE to publish XXX under the Creative Commons Attribution License (CCAL) CC BY 4.0 (http://creativecommons.org/licenses/by/4.0/ ). Please be aware that this license allows unrestricted use and distribution, even commercially, by third parties. Please reply and provide explicit written permission to publish XXX under a CC BY license and complete the attached form.”

Reviewers' comments:

Reviewer's Responses to Questions

**Comments to the Author**

1. Is the manuscript technically sound, and do the data support the conclusions?

Reviewer #1: Yes

Reviewer #2: Yes

2. Has the statistical analysis been performed appropriately and rigorously? 

Reviewer #1: Yes

Reviewer #2: Yes

3. Have the authors made all data underlying the findings in their manuscript fully available?

Reviewer #1: Yes

Reviewer #2: Yes

4. Is the manuscript presented in an intelligible fashion and written in standard English?

Reviewer #1: Yes

Reviewer #2: Yes

5. Review Comments to the Author

Reviewer #1: McCorriston and colleagues ms titled “South Arabia’s prehistoric monument landscape show social resilience to climate change” is an interesting addition to scholarly bibliography deserving further discussion. Built upon previous investigation by several of the authors, this paper presents results of original unpublished research: a multivariate analysis of 371 archaeological monuments from the Dhofar region of Oman. The main aim is to assess environmental and cultural factors influence on the location of different type of sites and their construction in a long durée perspective. The authors conclude that older monuments were built by larger groups during the Holocene Humid periods, but posterior aridification led to changes in the groups’ size resulting in changes from episodic to accretion construction techniques.

The Introduction gives an overview of the topics addressed and the authors state that they want to assess how measurable changes in megalithic monuments location and construction relate to social resilience of shifting demographics in the arid landscape of Dhofar region. Under the Materials and methods section, the authors present information concerning “Monument survey methods”, “Field documentation methods”, “Environmental data analysis and machine learning”, “Bootstrap aggregating”, “Environmental data analysis and regularized multinomial logistic regression”, “Collocation analysis”, “Technological and labor data analysis”. These sub-sections are clear and well-presented showing an important multivariate analysis accompanied by proper statistical testing. The Results are presented according to:

i) environmental factors influencing monument placement, showing spatial patterning (near to springs, low topographic positions, near to routes, and to higher vegetation biomass), some patterning of type of monuments in relation to environmental differences, and labour input as total stone volume;

ii) spatiotemporal relationships between monument, with clusters near water sources and different spatial relations according to type of monuments with interesting collocation results;

iii) technological and labour requirements for monument construction, lacking significant difference in total stone volume depending on the size of the largest stone, but with single-episode constructions being correlated with larger volume and accretive constructions with smaller volumes. Some interesting chronological patterns were observed.

Finally, the Discussion starts with a critic on the lack of holistic perspectives in the study of desert monuments, and a discussion on the relevance of these type of monuments in past pastoralist societies. The authors emphasize that the Platforms >> HCT shift (respectively, near to and further from springs) coincide with major aridification, which is related to subsistence changes (cattle < goat). It is discussed that prior monument placement was considered mostly to construct monuments far (and not closer) to preceding sites, and that changes in volume and size of stones indicates different investment in construction. It is also argued that HCT would relate to smaller mobile groups as a consequence of increasing aridity, triliths with increasingly smaller dispersed groups in a hyper-aridity environment, and Boat graves to an increased monsoon moisture after an arid period – these changes would represent new socio-cultural strategies in the dead-living relational world.

The ms makes a good case for the existence of associations between climatic and environmental factors but also social aspects. All data presented is accompanied by robust statistical tests and the main conclusions are properly presented and based on evidence or appropriate reasoning. Used bibliography is complete and updated. The figures and tables accompanying the text and as supplementary material are well prepared and informative, and data is freely available in tDAR. Overall, this ms is publishable, highly citable and scientific sound. Congratulations to the authors on this very interesting study.

On a minor note, please uniformize the use of “BP” and “ya” between the abstract, text and table 2. Finally, I would like to make a few comments and suggestions whose only objective is to contribute to making this ms clearer and stronger:

• “Monument survey methods” - it is not completely clear to the reader what is new concerning this sub-section and what was already published in previous papers of the team.

• Chronology - it should not deeply impact the argumentation, but I advise adding a more adequate and in-depth discussion of chronology in this paper since this is a critical aspect directly involved in the discussion and conclusions. The tight timeline sequence for the different types of monuments needs clarification because broader chronologies are not completely in accordance if this ms is compared to other studies (e.g., McCorriston et al. 2014). It seems that only a small sample of absolute dates exist and this should be discussed concerning its possible impact/bias on the chronological aspects of the ms. For example: developing a bit more on the SCABs presumed relative chronology attribution because none of your sample (n = 23) is dated; clarify the chronology and reuse of some of these monuments (e.g., HCT reuse during the 3rd and 2nd millennium BP; triliths dated construction or reuse) and the influence this might have on your results discussion. Studies in other regions show multi-period use of SCABS and I wonder if these multi-periods “episodic” use of monuments is as relevant in the samples presented in this study.

• Construction techniques – Supplementary material shows that all platform and HCT are episodic, one SCAB is accretive and the rest episodic, and all trilith and boat are accretive. Is the differentiation between “episodic” and “accretive” monuments clear cut (e.g. lines 136-138 / 316-328)? Specially concerning episodic monuments, it is not uncommon for large late prehistory megalithic sites (e.g., Western Europe alignments and cromlechs but also dolmens) to show multi-period modifications/additions and long separated uses. I understand what is the authors reasoning, but if there can be issues in properly understanding building (and use) phases it is better to mention that somewhere and be clear in relation to which were the variables considered to separate episodic from accretive constructions beyond monument typology (if any) in the text itself.

• Monuments description and sample size - which of the monuments under analysis were subject to archaeological excavations? I suggest clarifying this issue and eventually adding this information in the supplementary data (e.g., ASOM Monument Data.xls), as well as a small paragraph discussing the sample size for the different monument types because large differences exist among types.

• General aspects – several recent papers look at multi-proxies to discuss the impact of aridification phasis in late prehistoric populations, including “First Monument builders”, for example in terms of subsistence changes, among others. I think the paper would benefit from inserting some of this information on the Introduction/Discussion appealing to a wider audience and enriching the ms. Adding some broader information on the study area archaeological record dynamics that accompany the monuments and subsistence changes discussed, if existing, would also improve the ms.

• Conclusion - the ms lacks a short conclusion summarizing the main results/interpretations and possibly future research directions.

Reviewer #2: Review: South Arabia’s Prehistoric Monument Landscape Shows Social Resilience to Climate Change

Introduction

The manuscript explores the role of prehistoric monuments (although I prefer not to use the term “prehistory” as it can be pejorative to other communities who have history but lacks the means to write about them) in South Arabia, particularly in the Dhofar region of Oman, as adaptive social strategies to mitigate the effects of climate change. The authors utilize an interdisciplinary approach, combining archaeological evidence, GIS-based environmental analysis, and machine learning techniques to analyze 371 monuments and their environmental and cultural contexts over a span of 7,000 years. This study is highly innovative and contributes to our understanding of human-environment interactions in arid regions. While the manuscript is robust and demonstrates methodological rigor, certain aspects can be improved to enhance clarity, accessibility, and broader applicability.

The manuscript’s primary strength lies in its integration of diverse methodologies to examine the relationship between environmental change and human adaptation. By employing multivariate analysis and machine learning techniques alongside archaeological field data, the authors construct a comprehensive framework that links monument distribution and construction to socio-environmental factors. Additionally, the manuscript’s temporal scope provides a long-term perspective on the role of monuments in illustrating social resilience. The use of a robust dataset comprising 371 monuments, along with environmental variables such as proximity to springs, vegetation cover, and fog density, adds empirical weight to the conclusions. Moreover, the authors’ insights into labor investment and construction technology offer valuable contributions to the field of archaeology and resilience studies.

COMMENTS ON THE CONTENT

Clarity in Writing and Structure (revise only if targeting a general audience)

One significant challenge is the manuscript’s dense and technical writing style, which may limit its accessibility to a broader audience, including non-specialist readers. For instance, the methods section, while comprehensive, uses jargon and assumes familiarity with machine learning concepts and GIS-based analyses. Simplification and clearer explanations are necessary to make the manuscript more approachable.

To address this, the authors should consider restructuring the manuscript to include brief explanations of technical terms and methodologies. For example, while the use of bootstrap aggregating in machine learning is commendable, a layperson explanation of its purpose and advantages—such as reducing variance and avoiding overfitting—would greatly enhance reader comprehension. Similarly, the concept of accretive versus episodic construction is critical to the study’s findings but is introduced with minimal context. Providing more background on how these terms apply to archaeological practices would aid understanding.

Methodological Details

Although the manuscript demonstrates methodological rigor, certain details are either missing or underexplained, making it difficult for other researchers to reproduce the study. For instance, the criteria for categorizing monuments into episodic and accretive constructions are not fully articulated. Readers need a clearer understanding of how these classifications were determined based on archaeological evidence.

Moreover, the parameters used in regularized logistic regression and other statistical techniques require further justification. Why were specific thresholds or values chosen, and how do they align with previous studies? For example, the authors mention using a one-standard-error rule to select the optimal model for logistic regression but do not explain why this approach was preferred over alternatives. Expanding on these choices would strengthen the methodological transparency of the study.

Environmental and Cultural Context

While the manuscript effectively addresses environmental influences on monument placement, the cultural and social dimensions of these structures are underexplored. Monuments are described primarily as markers of social collectivity and resilience, but their deeper cultural significance is not sufficiently elaborated. How did these structures function within the social networks of ancient pastoralists? What cultural or symbolic meanings might they have conveyed?

To expand the discussion, the authors could incorporate ethnographic analogs or comparisons with other regions where pastoralist communities use monuments to establish social ties or mark territorial boundaries (perhaps in another paper). Such parallels would provide a more holistic understanding of the interplay between environment, culture, and resilience. Additionally, the authors could explore the potential spiritual or ritualistic functions of these monuments, as these aspects often play a critical role in their construction and placement.

Statistical Analysis and Visualizations

The presentation of statistical results is another area that requires attention. The figures and tables, while informative, are dense and challenging to interpret. For instance, Figure 5, which presents the coefficients from regularized logistic regression, is not easily understandable without additional context or explanation. Readers unfamiliar with this statistical technique may struggle to grasp its implications.

To improve accessibility, the authors should consider simplifying the visualizations or providing annotated versions that guide readers through the key findings. Partial dependence plots and other outputs could be accompanied by textual explanations that summarize the main takeaways. Additionally, including a brief tutorial or supplementary material on how these statistical techniques were implemented would be a valuable resource for readers.

Theoretical Framework

The concept of “social resilience” is central to the manuscript but is not clearly defined. While the authors cite resilience theory, they do not sufficiently engage with its broader theoretical underpinnings. What specific dimensions of resilience are being addressed, and how do they relate to the archaeological record? More importantly, what are there any parallels in other parts of the world?

A more explicit connection to resilience frameworks, such as coping, adaptive, and transformative capacities, would ground the study’s findings in a larger theoretical context. For example, the authors could discuss how the construction of monuments reflects adaptive strategies to environmental stressors or how these structures facilitated transformative social changes over millennia.

Collocation Analysis

The results of the collocation analysis, which show minimal spatial overlap between successive monument types, challenge the initial hypothesis that later monuments would cluster near earlier ones. This finding is intriguing but underexplored in the discussion. Why might monument builders have avoided earlier sites? Could this reflect shifts in settlement patterns, changes in social organization, or symbolic preferences for new locations?

The authors might want expand their interpretation of these results, considering alternative explanations and their broader implications for understanding monument landscapes. Additionally, they could discuss how these patterns compare to monument-building practices in other arid regions, thereby situating their findings within a global context.

Global Relevance

The issues identified in this manuscript—such as the interplay between environmental constraints, cultural practices, and technological adaptations—are not unique to South Arabia. Similar challenges and opportunities can be observed in monument landscapes worldwide. For example:

• Sub-Saharan Africa: Megalithic monuments, such as those in Ethiopia (Gedeo landscapes, perhaps), also reflect social strategies for resilience in arid and semi-arid environments. Like the Dhofar monuments, these structures often mark significant social or ritual spaces and are located in environmentally strategic areas.

• The Andes, South America: In regions such as the Peruvian highlands, ancient agricultural terraces and ritual spaces demonstrate how past societies adapted to environmental stressors. These structures also reflect communal labor investments and social networks, paralleling the episodic construction of Dhofar’s platforms.

• Atlas and Anti-Atlas Mountains (Morocco): The terraced agricultural landscapes in Morocco’s Atlas Mountains provide a striking example of how past and present societies have managed water and soil resources in challenging environments. These terraces, much like the Dhofar monuments, represent significant communal labor investments and illustrate the interplay between environmental adaptation and social organization.

By framing the findings in a comparative context, the authors could enhance the manuscript’s relevance to global discussions about resilience and human adaptation. Highlighting these parallels would also underscore the universality of certain adaptive strategies while respecting the unique cultural contexts of each region.

RECOMMENDATION

The manuscript presents a groundbreaking study that highlights the role of monuments as adaptive strategies for social resilience in arid environments. Its multidisciplinary approach and robust dataset make significant contributions to the fields of archaeology and resilience studies. However, revisions are needed to enhance clarity, methodological transparency, and theoretical depth. Moreover, positioning the findings within a global comparative framework would enrich the study’s contributions and demonstrate its broader applicability.

By addressing these issues, the authors can ensure that their findings reach a broader audience and have a lasting impact on both archaeological research and interdisciplinary studies of human-environment interactions. The manuscript is a strong candidate for publication in PLOS ONE following minor revisions.

6. PLOS authors have the option to publish the peer review history of their article (what does this mean? ). If published, this will include your full peer review and any attached files.

**Do you want your identity to be public for this peer review?** For information about this choice, including consent withdrawal, please see our Privacy Policy .

Reviewer #1: No

Reviewer #2: No

---

## [Author Response · Author response to Decision Letter 1]

11 Mar 2025

Response to Reviewers PONE-D-24-34740

Dear Editor Gordin and Reviewers,

Thank you for managing review of our manuscript South Arabia’s prehistoric monument landscape shows social resilience to climate change, submitted to PLOS ONE 13 August 2024, manuscript number PONE-D-24-34740. In returning revisions, we have paid close attention to your instructions and to the suggestions of reviewers, for which we are truly grateful. Per your request, we are uploading 1) a marked-up copy that highlights changes (“Revised Manuscript with Track Changes”) and 2) an unmarked version of our revised paper (“manuscript”).

Below please find a point-by-point response, first to your editor’s letter and next, to each reviewer. Line numbers refer to a version with TRACKED CHANGES VISIBLE.

Editor’s Letter

#1. We have followed PLOS ONE’s style requirements as indicated at the online locations.

#2. You mention that the ‘Funding Information’ and ‘Financial Disclosure’ sections do not match. In our resubmission, we tried to use the same statement for both sections, but the submission system still prints the first submission statement. The following is a revised statement for both “Funding Information” and “Financial Disclosure” sections. It has been edited from our original submission and produced here (italics)

This research has been funded by the US National Science Foundation under the Coupled Human - Natural Systems Large Grants (CNH-L 1617185) (PI-JMcC); and the US National Science Foundation Human Social Dynamics Program (DHB 0624268) (PI-JMcC); funds contributing to field work came from a National Geographic Explorer Grant # EC-44704R-18 (PI-AFB) and the Ministry of Heritage and Tourism, Sultanate of Oman (J.McC, A.A.Al-K, A.Al-M). The funders had no role in study design, data collection and analysis, decision to publish, or preparation of the manuscript.

#3. This work was not previously peer reviewed, nor published in whole or in part in the peer-reviewed literature. Our analysis is new, as are the data. Exceptions are the open access oxygen isotopic data in Figure 2 (obtained from Fleitmann et al. 2003, 2007 [54]). Some excavated tombs have been individually documented in a (non-peer reviewed) book we cite (McCorriston 2023) [55] where the overlap is some radiometric dates appearing in Figure 2 and described in Supplemental Table 3. Figure 2 also juxtaposes the published Bayesian posteriors from McCorriston & Dye 2020 [38] with other climate and radiometric data, thereby providing a new paleoclimate timeline in the top register of Figure 2. It is normal and innovative to synthesize data from multiple sources—some already published—to address a new focus with a new analytical approach and new data, as we have done here. There are many precedents for compiling paleodata from different open sources and researchers to arrive at new analytical conclusions (e.g., Petraglia et al. 2020 [26], Lézine et al. 2010 [24], 2017 [44]), including previous PLOS ONE publications (e.g., Crassard et al. 2020 https://doi.org/10.1371/journal.pone.0236314 )

#4. We have included an ethics statement in the ‘Methods’ section. Our study involved field research (overseas, international). It did NOT include human participants, human specimens or tissue, vertebrate animals or cephalopods, vertebrate embryos or tissues. Our statement reads: (italics) (Lines 213-215)

 In conformity with ethical standards, we conducted all field research under (un-numbered) permits issued annually through 5-year agreements signed between Ministry of Heritage and Tourism, Sultanate of Oman and Joy McCorriston (2007-2013, 2017-2021).

#5. All data are included in the manuscript with Supplemental Files and placed (also) in an online data repository (tDAR-the Digital Archaeological Record. Open data access will occur with publication, without restrictions.

For tDAR details, please see “Describe where the data may be found in full sentences…” section of the original submission statement. (We apologize for initial confusion over access at time of submission).

We note that in Comments to the Author #3, both reviewers found “Yes” to the question, “Have the authors made all data underlying the findings in their manuscript fully available?”

#6. Remove Identifying Data. This is confusing to us. There are no human research participants. Therefore there is no identifying data in the manuscript or Supporting Information files. Is it possible that column headers such as “Code,” “Distribution & Date,” place names (not people’s names) in the data, or the names of classes (of monuments) were erroneously flagged by an AI assistant during manuscript processing?

We have uploaded the data files again after careful consideration of #6.

#7. Maps have been reproduced using CC-BY-4.0 open datasets. Specifically, Mapzen terrain for the elevation raster, and geoBoundaries datasets for country (global ADM0) and region (ADM1) boundary polygons.

#8. We have reviewed all references to ensure they are complete, formatted correctly, and conform to your instructions regarding retractions. Per the reviewers’ comments, we added the following references to support clarifications and contextualization of our research:

31. Bocinsky and Kohler 2014

33. Burke et al. 2021

36. Burke, Riel-Salvatore, and Barton 2018 Caramanica et al. 2020 [32]

37. Chirikure et al. 2024

35. Clarke et al. 2016

64. Evans et al. 2014

27. Fedele 2008

54. Fleitman et al. 2003,2007

22. Howey, Palace, and McMichael 2016

23. Hudson et al. 2022

28. Uerpmann and Uerpmann 2008

Reviewer #1 (Review Comments to the Author)

[we address reviewer’s bulleted points as reviewer listed them]

“On a minor note...” --we have corrected for a uniform “BP” throughout abstract, text and table 2

• Monument survey methods. The reviewer seems to imply that the survey methods presented are not new, and this is indeed the case. For readers who may not have seen our prior publications, including those in a low circulation journal (J Oman Studies), we have 3 paragraphs summarizing how our data were collected and importantly (new), showcasing that the sampling over multiple years and projects nevertheless consistently collected all monument data within a defined viewshed (sampled area). We have edited the text for clarity on these points.

• Chronology. The reviewer advises adding “a more adequate and in-depts discussion of chronology...” We have added a section heading, “Chronological and stratigraphic methods” (Line 283) to discuss the absolute dates, the sample sizes, reuse of monuments, stratigraphic implications of samples, target events and dates. We have also amplified our discussion of the chronology presented here and its context within regional chronology (Hadramawt, Arabian Peninsula). Finally, in this expanded section, we also provide additional discussion of the dating of SCABS. (Lines 296-316)

Radiocarbon determinations come from excavated samples with varying potential to date the initial construction of a monument. Sometimes dated samples are from bone (human and faunal) and determine when an organism died. Other samples are carbon from charcoal, buried humus, or shell. We used stratigraphic associations to contextualize how the deposition of such samples relates to the actual event of monument construction. With multiple monuments of each type, we documented construction materials and techniques, preservation, use, and re-use, including subsequent burials and sacrifices, evidence of collapse, additions, and subsequently quarried stone [16, 55, 71, 72, 73]. The probability determinations in Figure 2 represent events prior or posterior to monument construction. Also, as a responsible reporting of results, we include a radiocarbon determination we think has unexplained error (D033-001). Another determination (D014-002) also lies outside our expected range. (For stratigraphic details see [14, 55]). More samples would surely refine and constrain our chronology, but we caution that excavation is a costly and sometimes unsuccessful path to obtain contextualized radiocarbon samples. Nevertheless, excavation remains the only viable strategy for a classification anchored in absolute dates.

Our classification and chronology largely correlate with observations across Arabia [3, 13, 15, 17, 18, 19, 20, 34, 39, 40, 74]. While multiple researchers have—like us—found episodes of re-use in existing monuments, we focus on the pattern of initial construction our studies of stone placements, arrangements, inclusions, additions, and radiocarbon dates indicate. As we documented through excavation, the reuse of HCT monuments occurred sometimes millennia after construction and disuse. This circumstance extended the HCT timeline in Hadramawt (pink dotted line in Figure 2). Elsewhere in Oman, Williams and Gregoricka [40] document re-visits and deposits subsequent to SCAB construction. Our SCAB sample (n=23) is undated, but we find some that share attributes of Neolithic Platforms, such as a standing stone and East-facing façade (e.g. D106-002); others are only some meters from Neolithic Platforms (e.g.,D027-001, D027-003). (S2-Table, S3-Table).

Finally, we have updated the caption for Figure 2 to clarify the reuse of monuments (evident from stratigraphic excavations)

The long dotted line for HCT reflects stratigraphic evidence of re-use centuries and millennia after primary interment

• Construction techniques. Reviewer requests further explanation for differentiation between “episodic” and “accretive” monuments. We have added manuscript text that describes which variables separate episodic from accretive, and we clarify that the classification refers only to initial construction, not to subsequent visits and modifications. (Lines 222-230, Lines 255-275). We also note that the removal of the file “Description of Monument Types.xlsx” per Editor’s Letter #6 (above) may have walled off reviewer’s access to this information in our submission.

Our terminology refers to the initial construction of monuments according to a structural-cognitive template. We recognize, as both stratigraphic examples and in theoretical concept, that monuments were re-visited, altered, additions made after substantial lapses in time, sometimes partially or fully dismantled, and appropriated—such is the role of monuments in both shaping and being shaped by social groups as literal touchstones for institutional thought [41, 67]. For example, through radiocarbon determinations, we documented instances of animal offerings placed into tombs opened and re-sealed a thousand years after a dated burial. This use is unrelated to initial construction, which was the target of our documentation through chronology and measurements.

For those that we could assign, we used the following visual criteria. Platforms have large blocks or upright slabs forming an outer base and an overall D-shaped or oval plan. Often the largest uprights form an eastern face. Platforms have a fill of smaller stones, and typically also have a (sometimes worked) standing stone set further to the east. (Excavations inside show that the uprights were supported by interior chock stones, in turn held in place by fill.) To balance and support stones and consolidate them into a free-standing, stable platform required rapid (i.e., episodic) initial construction. An HCT appears as a conical cairn 3-5 m across with a depressed center. This depression denotes an interior (usually collapsed) chamber made from upright or stacked slabs. Usually some of the chamber facing is still visible without excavation. Tomb (HCT) construction was surely also episodic, given the exigency of burial in a hot environment populated by scavengers and carnivores. A SCAB is also a conical structure, created by stacking boulders in concentric daises decreasing in diameter with the height of the monument. Triliths have clusters of 1-3 upright stones supported by a very low, oblong, gravel platform outlined by cobbles. It would be difficult to mistake this distinctive monument for any other type (Fig 1). Each trilith platform and its adjacent hearths is repeated along an alignment. We identified boat graves by their shape in plan and by the clustering of multiple, abutting, boat-shaped elements (Fig 1), each of which likely included a burial. In the cases of agglomerated triliths and boat graves, there is no construction characteristic that requires completion in a single episode. Triliths and boat graves could and sometimes did appear as isolated, unreplicated elements; this indicates that conglomerated examples were accretive, potentially completed over many years.

• Monument description and sample size. Per reviewer’s suggestion, we added a column (N “excavated”) in the S1-Table.xlsx (formerly “Monuments Data Table ST-1"). We also included a small paragraph discussing sample size and the inherent constraints on sampling. We have already explained that the regularized multinomial logistic regression “was preferred over machine learning techniques ....as it is more robust to the unequal numbers of observations of different monument types.” We note prior studies on the statistical significance of distributions across different landforms, and we suggest that our survey methods account for what has been established about these monuments in prior studies (Lines 702-709)

Large differences exist in the numbers of examples we recorded for each monument type. Prior experience in a comparable landscape (Hadramawt, southern Yemen) demonstrated that these numbers are not significantly explained by preservation conditions of different landforms. Moreover, platforms are significantly associated with lower elevations and HCT in higher elevations [58], both captured in our viewshed boundary in this research so that our monument counts would be relatively unaffected by terrain differences. Therefore other factors—such as period, cultural conventions, available labor—account for differences in the numbers of monuments of each type.

Per reviewer’s comment, we have added some broader information on the study area and the subsistence changes (Lines 76-82).

Albeit from few studies, Arabian faunal remains also suggest that cattle herders thrived in wetter environments in the early to mid-Holocene and adjusted their focus to different taxa according to available graze and browse as climates changed [27-29] so that cattle herders preferred wetter environments than today’s. In Dhofar’s steep environmental gradient, cattle herders today remain in the forested mountains and narrow plateau near the coast [30]. The arid interior is today sparsely populated by mobile goat and camel herders.

• Conclusion. Per reviewer’s suggestion, we have formulated and added to a short conclusion: (Lines 823-831)

Derived from the metrics of labor scheduling and location, and testable in new data sets, our empirical model theorizes eras of monument building as the exercises of social capacities to absorb risk in the dynamic climates and environments of the arid sub-tropics of Arabia. By decentering analysis from a single era or type of monument, we show how monument classes are proxies for social behavior of mobile, persistent pastoralists. Our model highlights a reliance on monuments as touchstones of social belonging and as flexible technologies of social resilience in a changing world. While built on archaeological observations of monuments in Arabia, our model may be applicable and adaptable to assess social resilience in other regions, such as Saharan, Mongolian, or the high Andes.

Reviewer #2 (Review Comments to the Author)

• Methodological Details. Both reviewers requested criteria for categorizing monuments into episodic and accretive constructions (addressed per Reviewer 1 above).

The reviewer has asked for simpler explanations of the analytical methods. We have edited these sections, specifically the methods on ML and regularized regression, reduced the use of jargon and removed non-essential technical explanation. We have provided further explanation or examples for unavoidable technical terms. We hope this makes the manuscript more readable and accessible to a range of audiences.

• Environmental and Cultural Context. While we recognize and echo the reviewer’s interest in the deeper cultural s

---

## [Decision Letter · Decision Letter 1]

10 Apr 2025

South Arabia’s prehistoric monument landscape shows social resilience to climate change

PONE-D-24-34740R1

Dear Dr. McCorriston,

We’re pleased to inform you that your manuscript has been judged scientifically suitable for publication and will be formally accepted for publication once it meets all outstanding technical requirements.

Kind regards,

Shai Gordin, Ph.D.

Academic Editor

PLOS ONE

Additional Editor Comments (optional):

Reviewers' comments:

Reviewer's Responses to Questions

**Comments to the Author**

1. If the authors have adequately addressed your comments raised in a previous round of review and you feel that this manuscript is now acceptable for publication, you may indicate that here to bypass the “Comments to the Author” section, enter your conflict of interest statement in the “Confidential to Editor” section, and submit your "Accept" recommendation.

Reviewer #1: All comments have been addressed

Reviewer #2: All comments have been addressed

2. Is the manuscript technically sound, and do the data support the conclusions?

Reviewer #1: Yes

Reviewer #2: Yes

3. Has the statistical analysis been performed appropriately and rigorously? 

Reviewer #1: Yes

Reviewer #2: Yes

4. Have the authors made all data underlying the findings in their manuscript fully available?

Reviewer #1: Yes

Reviewer #2: Yes

5. Is the manuscript presented in an intelligible fashion and written in standard English?

Reviewer #1: Yes

Reviewer #2: Yes

6. Review Comments to the Author

Reviewer #1: (No Response)

Reviewer #2: Thank you for addressing my initial comments. This ms looks good and would be a great contribution to archaeological studies on climate change.

7. PLOS authors have the option to publish the peer review history of their article (what does this mean? ). If published, this will include your full peer review and any attached files.

**Do you want your identity to be public for this peer review?** For information about this choice, including consent withdrawal, please see our Privacy Policy .

Reviewer #1: No

Reviewer #2: No

---

## [Editor Report · Acceptance letter]

PONE-D-24-34740R1

PLOS ONE

Dear Dr. McCorriston,

I'm pleased to inform you that your manuscript has been deemed suitable for publication in PLOS ONE. Congratulations! Your manuscript is now being handed over to our production team.

Kind regards,

on behalf of

Dr. Shai Gordin

Academic Editor

PLOS ONE